# RAIGen: Rare Attribute Identification in Text-to-Image Generative Models

**Silpa Vadakkeeveetil Sreelatha** [1]   **Dan Wang** [2]   **Serge Belongie** [3]   **Muhammad Awais** [1]   **Anjan Dutta** [1]

## Abstract

Text-to-image diffusion models achieve impressive generation quality but inherit and amplify training-data biases, skewing coverage of semantic attributes. Prior work addresses this in two ways. Closed-set approaches mitigate biases in predefined fairness categories (e.g., gender, race), assuming socially salient minority attributes are known a priori. Open-set approaches frame the task as bias identification, highlighting majority attributes that dominate outputs. Both overlook a complementary task: uncovering rare or minority features underrepresented in the data distribution (social, cultural, or stylistic) yet still encoded in model representations. We introduce RAIGen, the first framework, to our knowledge, for label-free rare-attribute discovery in diffusion models, requiring no predefined minority categories. RAIGen leverages Matryoshka Sparse Autoencoders and a novel minority metric combining neuron activation frequency with semantic distinctiveness to identify interpretable neurons whose top-activating images reveal underrepresented attributes. Experiments show RAIGen discovers attributes beyond fixed fairness categories in Stable Diffusion, scales to larger models such as SDXL, supports systematic auditing across architectures, and enables targeted amplification of rare attributes during generation. The project page is available at `https://vssilpa.github.io/RAIGen_webpage/`.

## 1. Introduction

Text-to-image (T2I) diffusion models such as Stable Diffusion have revolutionized image generation by producing high-fidelity visuals from natural language prompts (Podell et al., 2024; Rombach et al., 2022). However, these models not only reflect biases from their training data (Luccioni et al., 2023; Perera & Patel, 2023), but can also amplify them during generation, reinforcing societal stereotypes and inequalities if left unaddressed (Seshadri et al., 2024). For instance, despite near-parity in the LAION-5B dataset for occupations like "teacher", generated samples remain heavily gender-skewed (Friedrich et al., 2023). Such disparities reduce semantic coverage and raise concerns about fairness and real-world deployment.

Several approaches counteract biases in T2I generative models by rebalancing or diversifying their outputs (Chuang et al., 2023; Ni et al., 2024; Shen et al., 2024; Li et al., 2024). While effective for predefined categories like gender or race, they often overlook forms of underrepresentation, such as physical traits, cultural symbols, or stylistic variations, essential for semantic diversity and faithful generation. Open-set bias detection (D'Incà et al., 2024) broadens auditing but mainly identifies the majority attributes, and the surfaced attributes are largely dictated by inductive biases of external world models. Suppressing such majority features does not amplify the underrepresented ones, overlooking the critical task of identifying *minority attributes*: semantic factors encoded in the model's internal representations, but consistently underexpressed. We validate this empirically in Appendix G.1, showing that suppression of dominant attributes reduces their presence but reallocates probability mass unevenly across minority groups. This observation reinforces the need for systematic discovery of rare attributes as a prerequisite for comprehensive auditing and mitigation.

We introduce **RAIGen**, the first label-free framework for rare attribute identification in diffusion models where no predefined minority labels are required. Rather than identifying all possible underrepresented attributes, RAIGen targets those that are already encoded in the internal representations of the model but are systematically underexpressed during generation. These attributes are not hallucinated or externally defined, but emerge directly from the learned feature space of the model. By shifting the focus from majority identification to the structured discovery of these suppressed features, RAIGen enables more comprehensive auditing and understanding of representational gaps in generative models.

Identifying minority attributes requires access to the internal factors of variation learned by diffusion models. How-

---

[1]University of Surrey [2]University of California, San Diego [3]University of Copenhagen. Correspondence to: Silpa Vadakkeeveetil Sreelatha <s.vadakkeeveetilsreelatha@surrey.ac.uk>.

*Proceedings of the 43rd International Conference on Machine Learning*, Seoul, South Korea. PMLR 306, 2026. Copyright 2026 by the author(s).

ever, these representations are often entangled and uninterpretable. To address this, we require a mechanism that maps entangled internal representations into semantically interpretable features, a role effectively realized by Sparse Autoencoders (SAEs) (Kim et al., 2025). We specifically adopt Matryoshka Sparse Autoencoders (MSAEs), which have shown strong interpretability in vision-language models via hierarchical semantic decomposition (Pach et al., 2025; Zaigrajew et al., 2025). While finer, more overcomplete levels could in principle better isolate minority attributes hidden within broader concepts, in practice they often fragment a single concept into many localized part-features, inflating the search space and yielding brittle or spurious rare attributes that are not stable, human-interpretable factors. We thus focus on the coarsest MSAE level, whose features are typically semantically and spatially coherent, producing more reliable attribute hypotheses.

Given these coarse features, a natural heuristic for minority attribute discovery is the activation frequency of MSAE neurons. We validate this approach in a controlled toy setting with known rare factors, where the least frequently activated neurons consistently align with the injected minority attributes (Section 5.1), establishing frequency as a reliable proxy for rarity. However, in a real-world setting, low activation alone may correspond to non-semantic or noisy neurons. To address this, we incorporate semantic distinctiveness, measuring how far the neuron's top-activating samples lie from the dataset's average semantic representation. Our final minority score combines rarity and distinctiveness, prioritizing neurons that are both infrequent and semantically separated. MSAEs further support verification via top-activating samples and spatial heatmaps, enabling direct inspection of neuron-level attributes.

The key contributions of this work are as follows: ❶ To the best of our knowledge, we introduce the first framework for rare attribute identification in diffusion models, extending bias analysis from predefined fairness categories or majority-dominant features to the systematic identification of underrepresented attributes encoded in model representations. ❷ We propose a simple, yet effective, minority metric that combines neuron activation frequency with semantic distinctiveness, forming the basis of RAIGen. ❸ We show that RAIGen reveals attributes beyond fairness categories, enables auditing across multiple diffusion architectures (Stable Diffusion 1.5, 2, XL, FLUX.1-schnell), and supports amplification via lightweight prompt interventions.

## 2. Related Work

T2I generation has significantly advanced generative AI, enabling the creation of highly realistic images from textual prompts (Ho et al., 2020; Ramesh et al., 2022; Rombach et al., 2022), but they also inherit and amplify the biases in

their training data (Cho et al., 2023; Luccioni et al., 2023).

**Bias Mitigation in Diffusion Models:** Several methods mitigate biases in diffusion models such as Stable Diffusion. (Chuang et al., 2023) learn projection matrices on text embeddings aligned with fairness attributes; (Friedrich et al., 2023) and (Parihar et al., 2024) use classifier-free guidance to steer generations without retraining; and (Li et al., 2024; Vadakkeeveetil Sreelatha et al., 2025) introduce learnable modules on bottleneck representations to enforce responsible concepts while preserving semantic alignment. All assume the target attributes are known a priori. In contrast, we seek to uncover minority attributes that are already encoded in the model but systematically underexpressed, beyond any predefined fairness category.

**Unknown bias identification:** Bias auditing in text-to-image models has shifted from mitigating predefined categories to identifying previously unknown biases. Open-set bias detection emphasizes uncovering such biases without relying on predefined labels. (D'Incà et al., 2024) proposes a framework to automatically identify biases in generative models by leveraging large language models to suggest potential bias attributes, generating synthetic images, and applying visual question answering to rank the prevalence of these biases. However, such methods mainly surface majority attributes that dominate generations, revealing overrepresentation but not underrepresentation. We address this gap by shifting the focus from identifying dominant biases to discovering minority attributes that are suppressed.

**Interpretability with Sparse Autoencoders:** Sparse autoencoders (SAEs) have become useful tools for interpreting generative models by mapping intermediate activations to human-interpretable concepts, enabling concept steering, suppression, and unlearning (Kim et al., 2025; Surkov et al., 2025; Tinaz et al., 2025; Cywiński & Deja, 2025). Recent work further extends SAEs to hierarchical representations, such as Matryoshka SAEs for coarse-to-fine interpretability in CLIP (Pach et al., 2025). In responsible generation, DiffLens (Shi et al., 2025) and SAeUron (Cywiński & Deja, 2025) intervene on SAE features associated with predefined sensitive attributes or target concepts. In contrast, RAIGen tackles the upstream open-set problem of identifying which attributes are encoded but suppressed, without assuming categories in advance.

## 3. Preliminaries

**Diffusion Models:** Diffusion models (Sohl-Dickstein et al., 2015; Ho et al., 2020; Song & Ermon, 2019) synthesize data by learning to reverse a forward process that progressively adds Gaussian noise to a clean sample $\mathbf{x}_0 \sim p_{\text{data}}$ according to a variance schedule, yielding $\mathbf{x}_T \sim \mathcal{N}(\mathbf{0}, \mathbf{I})$ as $t \to T$. A neural network $\boldsymbol{\epsilon}_\theta(\mathbf{x}_t, t)$ learns to predict the

added noise, defining a denoising transition at each step. At inference, generation starts from noise and recursively applies the learned reverse process to produce a clean sample. T2I models such as Stable Diffusion (Rombach et al., 2022) extend this framework by operating in a compressed latent space where they condition on text embeddings from a language encoder, aligning generated images with language.

**Sparse Autoencoders (SAEs):** Sparse autoencoders aim to decompose input representations $\mathbf{r} \in \mathbb{R}^n$ into a set of latent features $\mathbf{z} = \{z_1, \ldots, z_d\} \in \mathbb{R}^d$ that are both overcomplete $(d \gg n)$ and sparse, thereby encouraging interpretability and disentanglement of concepts. The encoder-decoder architecture is defined as:

$$\mathbf{z} = \mathrm{ReLU}(W_{\mathrm{enc}}(\mathbf{r} - \mathbf{b}_{\mathrm{pre}}) + \mathbf{b}_{\mathrm{enc}}), \quad \hat{\mathbf{r}} = W_{\mathrm{dec}}\mathbf{z} + \mathbf{b}_{\mathrm{pre}},$$

where $W_{\mathrm{enc}} \in \mathbb{R}^{d \times n}$, $W_{\mathrm{dec}} \in \mathbb{R}^{n \times d}$, and $\mathbf{b}_{\mathrm{enc}} \in \mathbb{R}^d$, $\mathbf{b}_{\mathrm{pre}} \in \mathbb{R}^n$ are learnable parameters. The model is trained to minimize the reconstruction loss $\mathcal{L}_{\mathrm{SAE}} = \|\mathbf{r} - \hat{\mathbf{r}}\|_2^2$, while enforcing sparsity on $\mathbf{z}$. Sparsity is imposed either through $\ell_1$ penalties on $\mathbf{z}$ (Bricken et al., 2023), which can cause activation shrinkage (Rajamanoharan et al., 2024), or by hard selection of the top-$k$ coordinates per input (Gao et al., 2025), which enforces exact sparsity but fixes the number of active units. BatchTopK (Bussmann et al., 2025) modifies this by flattening all activations in a batch into a single vector and retaining the largest $k \times B$ entries (for batch size $B$), allowing the number of active features to vary across samples while maintaining a global sparsity constraint.

**Matryoshka Sparse Autoencoders (MSAEs):** MSAEs extend SAEs by training under multiple sparsity constraints at once, following the idea of Matryoshka representation learning (Kusupati et al., 2022). Instead of selecting a single sparsity level $k$, the model applies a family of Top-$k$ operators with increasing levels $\{k_1, k_2, \ldots, k_f\}$, where $k_1 < k_2 < \cdots < k_f = d$, forming a nested budget: $k_1$ active neurons at the first level, then $(k_2 - k_1)$ more at the next, and so forth, up to $d$. For an input $\mathbf{r}$, the encoder produces multiple sparse codes and reconstructions for each level as follows:

$$\mathbf{z}^{(k_i)} = \mathrm{ReLU}\big(\mathrm{Top}_{k_i}(W_{\mathrm{enc}}(\mathbf{r} - \mathbf{b}_{\mathrm{pre}}) + \mathbf{b}_{\mathrm{enc}})\big),$$
$$\hat{\mathbf{r}}^{(k_i)} = W_{\mathrm{dec}}\mathbf{z}^{(k_i)} + \mathbf{b}_{\mathrm{pre}}.$$

The training objective aggregates reconstruction losses across all levels,

$$\mathcal{L}_{\mathrm{MSAE}} = \sum_{i=1}^{f} \alpha_i \|\mathbf{r} - \hat{\mathbf{r}}^{(k_i)}\|_2^2, \tag{1}$$

with coefficients $\alpha_i$ weighting the contribution of each sparsity level. At inference, any $k_i$ can be probed to reveal features at varying granularities. This design produces a hierarchical representation, with coarse levels capturing broad semantics and finer levels encoding detailed attributes.

## 4. Methodology

We propose **RAIGen**, a framework for rare attribute identification in text-to-image diffusion models. An overview of the framework is illustrated in Figure 1.

### 4.1. Problem Formulation

Let $\mathcal{A} = \{a_1, a_2, \ldots, a_m\}$ denote the set of semantic attributes that may be expressed in the outputs of a conditional generative model, where $m \geq 2$. For instance, $\mathcal{A}$ could include {male, female, urban background, rural background, dark skin tone, …}. A conditional generative model is defined as $G : (\boldsymbol{\xi}, \mathbf{c}) \mapsto \mathbf{x}$, where $\boldsymbol{\xi} \sim \mathcal{N}(\mathbf{0}, \mathbf{I})$ is a latent variable, $\mathbf{c} \in \mathcal{C}$ is an external condition (for example, a text prompt), and $\mathbf{x}$ is the generated output. The model induces a conditional probability distribution over attribute values:

$$P_G(a_i \mid \mathbf{c}) = \Pr[A(\mathbf{x}) = a_i \mid \mathbf{c}], \quad i = 1, \ldots, m.$$

where $A(\mathbf{x})$ denotes the attribute value associated with the generated sample $\mathbf{x}$.

**Definition 1** (**Generative Bias**). A model $G$ exhibits *generative bias* (Ferrara, 2024; Huang & Huang, 2025) with respect to attribute set $\mathcal{A}$ under condition $\mathbf{c}$ if there exist $i \neq j$ such that

$$P_G(a_i \mid \mathbf{c}) \neq P_G(a_j \mid \mathbf{c}).$$

That is, the model assigns uneven probabilities to attribute values under identical conditions.

**Definition 2** (**Minority Attribute**). For a tolerance parameter $\epsilon > 0$, an attribute $a_j \in \mathcal{A}$ is a *minority attribute* under condition $\mathbf{c}$ if

$$0 < P_G(a_j \mid \mathbf{c}) \leq \min_{a_i \in \mathcal{A} \setminus \{a_j\}} P_G(a_i \mid \mathbf{c}) + \epsilon$$

This definition implies two conditions: (1) attributes with $P_G(a_j \mid \mathbf{c}) = 0$ are excluded, as they are not represented in the model's latent space; (2) minority attributes are defined with respect to the model's generative distribution rather than the raw training data, identifying features that are internally encoded but suppressed in outputs.

**Objective:** The discovery of minority attributes requires a mechanism that meets the following criteria: (1) It exposes the set of internal semantic concepts encoded by the generative model $G$. Formally, let $\mathbf{h}$ denote the representations extracted from $G$. We aim to utilize a feature decomposition operator $M$ that maps the representations into a set of sparse latent features $M(\mathbf{h}) = \mathbf{z} = \{z_1, \ldots, z_d\} \in \mathbb{R}^d$. Throughout, we use the term *neuron* to refer to individual latent feature $z_i$ in the sparse representation $\mathbf{z}$, each of which may capture a distinct semantic feature. (2) It assigns a quantitative score reflecting the degree of underrepresentation of

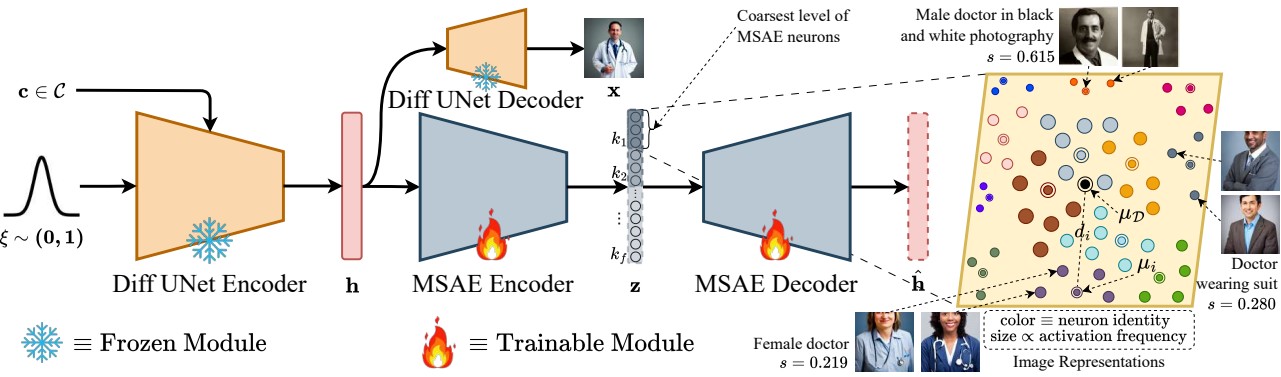

*Figure 1.* Overview of **RAIGen**. Diffusion representations (**h**) are decomposed by MSAE into interpretable features (**z**). A minority score ($s$), combining rarity and distinctiveness, ranks neurons or features to reveal minority attributes. Minority concepts are identified at the coarsest MSAE level (e.g., female doctor, doctor in suit), where size reflects activation frequency (smaller size = less frequent) and color denotes neuron identity.

each feature. We define a scoring function $s$ that assigns each latent feature $z_i$ a value $s(z_i) \in [0, 1]$, which yields the score vector $s(\mathbf{z}) = (s(z_1), \dots, s(z_d)) \in [0, 1]^d$. Each $s(z_i)$ captures both the rarity of feature $z_i$ under condition **c** and its semantic distinctiveness relative to other features in **z**.

This motivates RAIGen, which employs MSAEs to decompose the representations **h** into semantically meaningful features **z**, and introduces a novel *minority score $s(\mathbf{z})$* that integrates feature rarity with semantic distinctiveness. Together, these components enable the label-free discovery of minority attributes directly from the internal representations of diffusion models.

### 4.2. Rare Attribute Identification

**Feature Decomposition from Diffusion Representations:** To expose the internal concepts encoded by diffusion models, we train MSAE on intermediate representations extracted during reverse sampling. Given a T2I diffusion model $G$ and a prompt **c**, we extract bottleneck representations $\mathbf{h}_t \in \mathbb{R}^{h \times w \times n}$ at each denoising step $t$. These representations are inherently interpretable (Kwon et al., 2023), and (Kim et al., 2025) have shown that SAEs trained on them reveal high-level features. Following (Cywiński & Deja, 2025), we treat each spatial location in $\mathbf{h}_t$ as an $n$-dimensional training example, disregarding spatial coordinates. These vectors are then used to train a MSAE using Equation (1), yielding a hierarchy of sparse codes $\mathbf{z}^{(k_i)}$ that capture semantic structure at varying levels of granularity from broad concepts at coarse levels ($k_1$) to finer details at deeper levels ($k_f$).

For minority attribute discovery, we perform inference with MSAE by collecting representation-image pairs $\mathcal{D}_c = \left\{ (\mathbf{h}_t^{(j)}, \mathbf{x}^{(j)}) \right\}_{j=1}^N$, where $\mathbf{h}_t^{(j)} \in \mathbb{R}^{h \times w \times n}$ denotes bottleneck representation at a fixed denoising step $t$, and $\mathbf{x}^{(j)}$

is the corresponding generated image for a prompt $c$. In practice, we use the final timestep, where semantic information is most fully expressed. For simplicity, we omit both the sample index $j$ and the timestep index $t$, and write $(\mathbf{h}, \mathbf{x}) \in \mathcal{D}_c$ for an arbitrary pair. Each tensor **h** is flattened into $h \times w$ feature vectors, which are individually passed through the MSAE encoder following the training setup. For each MSAE neuron $z_i$, with $i = 1, \dots, d$ corresponding to a sparse latent feature, we define its activation on **h** as $z_i(\mathbf{h})$, obtained by averaging activations across spatial positions. These per-neuron activations form the basis for computing the minority score.

**Minority Score:** To quantify the degree to which each neuron encodes a minority attribute, we introduce the *Minority Score*, which balances two complementary signals: rarity of activation and semantic distinctiveness. Let $(\mathbf{h}, \mathbf{x}) \in \mathcal{D}_c$ be a diffusion representation-image pair, and $z_i(\mathbf{h})$ the activation of MSAE neuron $z_i$ as previously defined. We define the activation frequency as the proportion of samples where the neuron is active (i.e., has nonzero activation):

$$\nu_i = \frac{|\{(\mathbf{h}, \mathbf{x}) \in \mathcal{D}_c : z_i(\mathbf{h}) > 0\}|}{|\mathcal{D}_c|}. \quad (2)$$

This metric directly measures how often the neuron participates across the dataset $\mathcal{D}_c$, with rarer features corresponding to lower $\nu_i$. We empirically validate activation frequency as a rarity signal in a controlled toy experiment (Section 5.1), and find that the least frequently activated neurons are more likely to correspond to rare features. In a real-world setting, however, low $\nu_i$ can be noisy and may reflect uninterpretable activations, so we complement it with semantic distinctiveness.

We evaluate the semantic distinctiveness of each neuron by comparing its activation-weighted CLIP centroid to the global dataset centroid. Let $\text{CLIP}(\mathbf{x})$ denote the CLIP embedding of image **x**. The centroid $\mu_i$ for neuron $z_i$, and the

global centroid $\mu_{\mathcal{D}_c}$, are computed as:

$$\mu_i = \frac{\sum_{(\mathbf{h},\mathbf{x}) \in \mathcal{D}_c} z_i(\mathbf{h}) \, \text{CLIP}(\mathbf{x})}{\sum_{(\mathbf{h},\mathbf{x}) \in \mathcal{D}_c} z_i(\mathbf{h})},$$

$$\mu_{\mathcal{D}_c} = \frac{1}{|\mathcal{D}_c|} \sum_{(\mathbf{h},\mathbf{x}) \in \mathcal{D}_c} \text{CLIP}(\mathbf{x}). \qquad (3)$$

Semantic distinctiveness $d_i$ is then defined as the cosine distance between the two centroids. This metric ensures that the neuron centroid $\mu_i$ is dominated by its top-activating images, since activation strengths directly weight their contribution. In contrast, the global dataset centroid $\mu_{\mathcal{D}_c}$ represents the average semantics of the entire set of images in $\mathcal{D}_c$. The resulting distance $d_i$ thus measures how much the neuron's semantics deviate from the dataset's average semantic representation. Both $d_i$ and $\nu_i$ are min–max normalized to $[0, 1]$ for comparability. *Minority Score* is then defined as:

$$s(\mathbf{z}) = \mathbf{d} \odot (\mathbf{1} - \boldsymbol{\nu}) \qquad (4)$$

where $\mathbf{d} = (d_1, \ldots, d_d)$, $\boldsymbol{\nu} = (\nu_1, \ldots, \nu_d)$. This formulation assigns a high score to neurons that are both rarely active (low $\nu_i$) and semantically distinct (high $d_i$) relative to the dataset's average semantics. Intuitively, neurons with larger $s(z_i)$ are more likely to encode minority attributes, since they capture concepts that occur infrequently yet deviate substantially from dominant patterns [1]. We demonstrate in Appendix G.10 that combining activation frequency and semantic distinctiveness is necessary to reliably recover minority-associated neurons. Conversely, neurons with the lowest minority scores do not necessarily correspond to dominant attributes, since low values can also arise from noisy or undistinctive activations, as we explain in detail in Appendix G.3. Although the Minority Score can be computed across all MSAE neurons, we focus on the coarsest level $z(k_1)$, which captures broad, interpretable semantics. By restricting analysis to the top-$k_1$ codes, we prioritize high-level structure over low-level noise, leveraging the hierarchical design of MSAEs to expose global attributes more clearly than standard SAEs.

Minority concepts often appear redundantly across multiple neurons with similar activation patterns. To obtain a compact and diverse set, we use the neuron centroid $\mu_i$ (Eq. 3) as a representative of each neuron's semantics. Redundancy is assessed via pairwise cosine distances between centroids, which quantify similarity between neurons. We then iterate over neurons in descending order of *Minority Score*, retaining the current neuron in the final set and removing all others within a small fixed distance. This threshold, treated as a

---

[1]The framework also accommodates a prompt-independent setting: training the MSAE on representations pooled across a diverse prompt mixture yields globally underrepresented attributes spanning the entire generative distribution, while the Minority Score computation remains identical.

hyperparameter, controls semantic redundancy. The resulting set therefore contains distinct, interpretable, minority neurons ranked by underrepresentation. For interpretability, we visualize top-activating images with MSAE heatmaps and provide human-readable annotations via MLLMs.

We refer to RAIGen as *label-free* rather than fully unsupervised. The discovery pipeline: MSAE training, the Minority Score, and neuron ranking, requires no predefined minority categories or attribute labels, and operates purely on the diffusion model's internal representations. However, we do rely on a pretrained semantic prior, a CLIP-style image encoder used in Eq. 3 to define semantic distinctiveness, which provides semantic geometry but not attribute supervision.

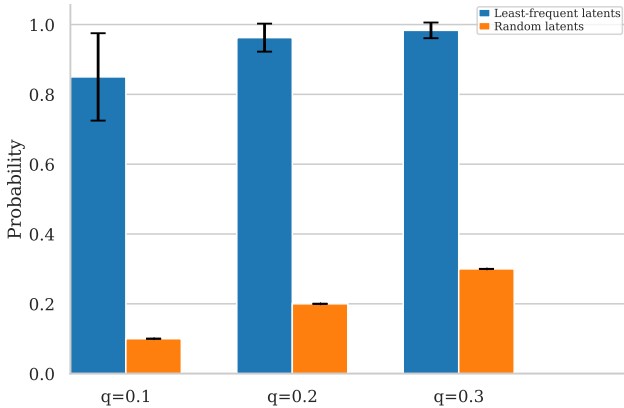

*Figure 2.* **Least-active latents preferentially map to rare ground-truth features in the toy setting.** The **blue bars** report $\mathbb{P}(\text{rare feature} \mid \text{latent} \in \text{least-active})$, i.e., the fraction of least-active latents whose matched feature is rare. The **orange bars** report the same probability for a random baseline, computed by sampling the same number of latents uniformly at random.

## 5. Experiments

We first validate activation frequency as a signal for rarity in a controlled toy experiment. We then qualitatively and quantitatively evaluate RAIGen-discovered minority attributes on Stable Diffusion v1.4 and SDXL, including a user study, and show that these attributes can be systematically amplified during generation. Appendix G.2 demonstrates that RAIGen enables auditing across diffusion architectures, and Appendix 5.5 shows that RAIGen extends to FLUX.1-schnell. Additional experiments are reported in Appendix G.

### 5.1. Activation Frequency as a Proxy for Rarity

In this section, we test the central heuristic underlying our minority metric: *latents with low activation frequency correspond to rare underlying factors*. To evaluate this claim in a setting with known ground truth, we replicate the hierarchical tree-structured toy data generation procedure of Bussmann et al. (2025), which produces a long-tailed dis-

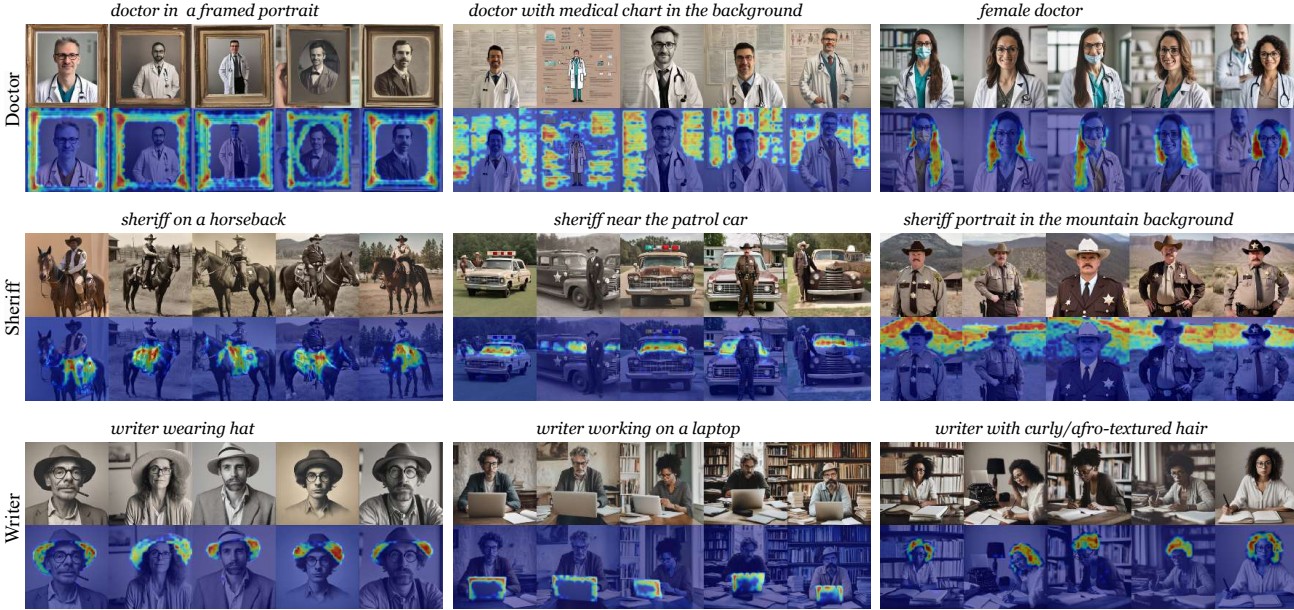

*Figure 3.* **WinoBias qualitative examples on SDXL.** Top-activating images and MSAE activation heatmaps for minority neurons discovered by RAIGen across three WinoBias profession prompts: *Doctor* (top), *Sheriff* (middle), and *Writer* (bottom). Labels above each group show generated language annotations for the corresponding neuron.

tribution of feature frequencies (Figure 6). We then train a Matryoshka SAE under their toy configuration and align learned latents to ground-truth features via a one-to-one Hungarian assignment computed from activation-similarity statistics, following Bussmann et al. (2025). Using this alignment, we assess (i) the relationship between each latent's firing rate and the empirical frequency of its matched ground-truth feature, and (ii) whether the least-active Matryoshka latents disproportionately map to the rarest ground-truth features. We repeat all experiments over 20 seeds.

For a quantile level $q \in \{0.1, 0.2, 0.3\}$, we operationalize rarity as follows. After matching latents to ground-truth features, we define (i) least-active latents as the bottom-$q$ fraction of matched latents ranked by firing rate, and (ii) rare features as the bottom-$q$ fraction of matched ground-truth features ranked by their true activation frequency. Figure 2 reports the conditional probability that a latent drawn from the least-active set is matched to a rare feature, $\mathbb{P}(\text{rare feature} \mid \text{least-active latent})$, and contrasts it with a random-latent baseline. For all three quantiles, the least-active set exhibits a substantially higher probability of mapping to rare features than random, indicating that low latent firing rate is informative of true feature rarity in this controlled setting. This is further supported by a strong monotonic association between latent firing rates and the frequencies of their matched ground-truth features, with mean Spearman correlation $\rho \approx 0.991$ across seeds. Taken together, these results suggest that activation frequency serves as a reliable proxy for rarity. We view this as a validation of

the heuristic under near-ideal feature–latent alignment; the ablations in Appendix G.10 show why frequency alone is insufficient in practice.

## 5.2. Minority Attributes Discovery

**Datasets:** We perform minority attribute discovery on WinoBias (Zhao et al., 2018) and COCO prompts (Lin et al., 2015) to ground our analysis in datasets that capture complementary aspects of bias and diversity. WinoBias provides controlled, occupation-based prompts that are widely used in fairness evaluations, making it well-suited for testing whether RAIGen can surface socially salient minority attributes such as gender imbalances in profession-related generations. In contrast, COCO offers a broad and diverse distribution of everyday scenes, and contexts, allowing us to evaluate whether RAIGen can uncover underrepresented attributes beyond fairness categories.

**Experimental setting:** We run minority attribute discovery on Stable Diffusion v1.4 (SD v1.4) (Rombach et al., 2022) and SDXL (Podell et al., 2024), a substantially larger and higher-capacity text-to-image model. For each prompt in WinoBias and COCO, we generate images and extract the model's bottleneck representations. We use these representations to train MSAE and then identify underrepresented attributes using the procedure described in Section 4. For interpretability, we annotate the resulting minority neurons with GPT-5.2. Additional experimental details are provided in Appendix E.

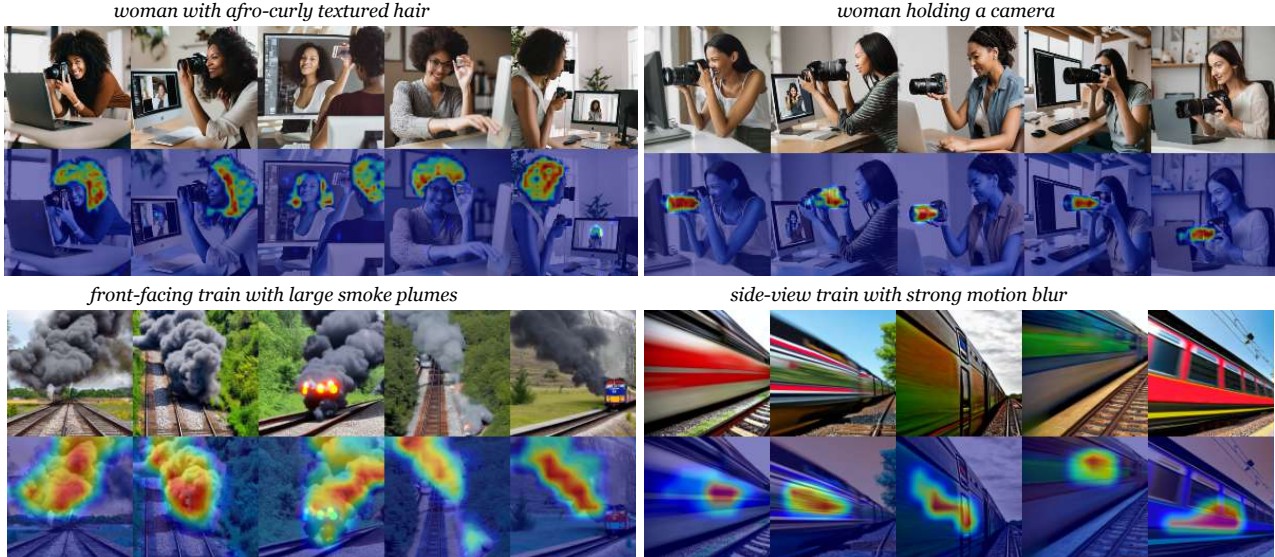

*Figure 4.* **COCO qualitative examples on SDXL and SD v1.4.** Top-activating images and MSAE activation heatmaps for minority neurons discovered by RAIGen for two COCO prompts: *"A woman taking a picture of herself in front of a desktop"* using SDXL (top row) and *"A train going down a track at full speed"* using SD v1.4 (bottom). Labels above each group indicate generated language annotation for the corresponding neuron.

We evaluate RAIGen using an Attribute Presence metric based on VQA-style attribute verification commonly used in text-to-image evaluation (Hu et al., 2023). For each discovered minority attribute, we convert its language annotation (Section 4) into an attribute query and apply it to every generated image. To mitigate evaluator-specific biases, we do not rely on a single vision-language model; instead, we query both Llama 4-Scout (AI, 2024) and Qwen3-VL-8B-Instruct (Bai et al., 2025) for each image–attribute pair and aggregate their binary outputs via majority vote. Attribute Presence is then defined as the fraction of images for which the ensemble predicts the attribute is present (lower values indicate stronger rarity). We compare RAIGen against Open-Bias (D'Incà et al., 2024), which targets majority attributes; while prior open-set approaches reveal what models tend to overproduce, RAIGen complements them by uncovering attributes the model systematically suppresses. Additional details on evaluation is in Appendix F.

**Quantitative results:** Table 1 reports Attribute Presence for Stable Diffusion v1.4 and SDXL on WinoBias (Prof) and COCO, comparing minority attributes surfaced by RAIGen to majority attributes discovered by OpenBias (D'Incà et al., 2024). Across both datasets, OpenBias attributes occur with high frequency, whereas RAIGen attributes appear substantially less often, confirming that RAIGen isolates features that are encoded but rarely expressed under standard sampling. Notably, RAIGen presence is slightly lower in SDXL than SD v1.4 on both WinoBias and COCO, suggesting that increased model capacity does not necessarily translate into higher expression of rare modes.

| Model | Approach | WinoBias($\downarrow$) | COCO ($\downarrow$) |
|---|---|---|---|
| SD v1.4 | OpenBias | 0.941 | 0.933 |
| | RAIGen | **0.205** | **0.220** |
| SDXL | OpenBias | 0.941 | 0.933 |
| | RAIGen | **0.194** | **0.199** |

*Table 1.* Attribute Presence for majority (OpenBias) and minority (RAIGen) attributes on WinoBias (Prof) and COCO. Lower values indicate stronger underrepresentation.

**Qualitative results:** To assess RAIGen qualitatively, we visualize top-activating images for discovered MSAE neurons with activation heatmaps, and the corresponding language annotations. Figure 3 shows WinoBias results on SDXL. RAIGen uncovers both socially salient minority attributes and non-fairness concepts: for example, it surfaces neurons corresponding to female doctor as well as presentation/context cues such as *doctor in a framed portrait* and *doctor with a medical chart in the background*.

Figure 4 presents COCO examples for two prompts with SDXL results in the first row and SD v1.4 results in the second row. Across both models, RAIGen surfaces coherent minority attributes that correspond to rarely expressed semantics, including appearance and object-centric cues (e.g., *afro-curly textured hair*, *a woman explicitly holding a camera*) as well as distinctive scene modes (e.g., *front-facing trains with large smoke plumes* or *side-view trains with pronounced motion blur*). Overall, these examples show that RAIGen uncovers underrepresented attributes beyond fair-

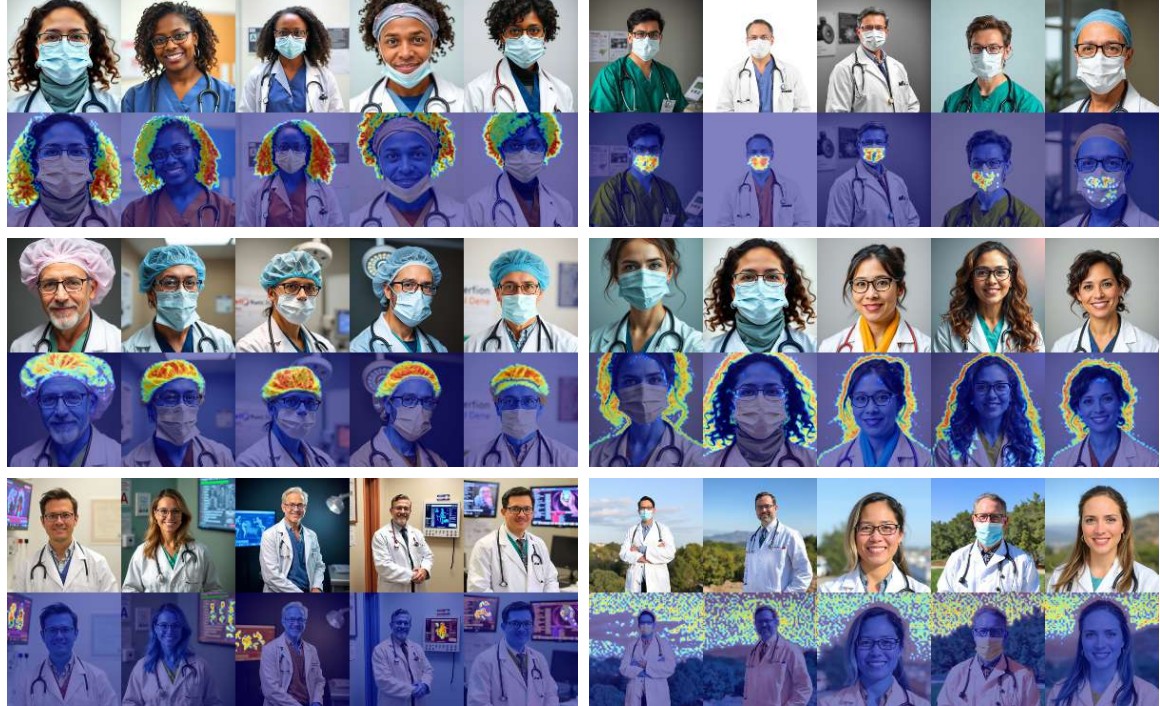

*Figure 5.* **RAIGen on FLUX.1-schnell (transformer-based diffusion).** Qualitative examples of minority neurons discovered for the prompt *"a photo of a doctor"*. For each neuron, we show its top-activating generated images and the corresponding activation heatmaps.

ness categories, spanning fine-grained stylistic, contextual, and compositional modes; importantly, these rare modes are detectable in both SDXL and SD v1.4 generations. Additional qualitative examples appear in Appendix H.

### 5.3. User study

*Table 2.* Human-estimated presence of RAIGen minority attributes (expected count out of 10 images; lower is rarer).

| Profession | Avg. Mean Presence ($\downarrow$) | 95% CI |
|---|---|---|
| Analyst | 1.35 | [1.03, 1.67] |
| CEO | 0.70 | [0.44, 0.96] |
| Doctor | 1.18 | [0.97, 1.39] |
| Salesperson | 1.45 | [0.99, 1.91] |
| Sheriff | 2.64 | [2.21, 3.07] |

To test whether RAIGen's minority attributes are systematically underexpressed in default generations, we ran a user study with 25 participants. We consider five WinoBias professions (ANALYST, CEO, DOCTOR, SALESPERSON, SHERIFF) and, for each profession, the top-6 minority attributes discovered by RAIGen. For each profession, participants viewed a participant-specific grid of 10 images randomly sampled from SD v1.4 and, for each attribute, answered: *"How many of the images in this grid contain this attribute?"* (integer response in $[0, 10]$). For each pro-

fession, we compute a participant-level presence score by averaging each participant's counts across the six attributes. We report the profession-level mean of these scores, with 95% confidence intervals across participants. Table 2 summarizes the results. Although RAIGen identifies attributes from intermediate representations, this study deliberately evaluates them at the image level: the perceptual ground truth is whether the attribute is visibly expressed in the generated outputs. This design directly tests whether RAIGen surfaces attributes that are not only internally encoded, but also meaningfully rare in the model's standard generations.

Across professions, participants report RAIGen attributes in fewer than 3 out of 10 images on average, providing direct human evidence that these attributes are rarely expressed under standard sampling despite being encoded in the model's representation space. The scarcity is most pronounced for CEO, where participants rarely observe the discovered attributes. Even for the profession with the highest presence, SHERIFF, the attributes appear only in a small minority of images. Overall, these results show that RAIGen's discovered attributes are not only interpretable, but also recognized by humans as minority attributes: they are perceptible when present, yet occur infrequently under standard sampling, making RAIGen a practical tool for auditing rare modes beyond the model's dominant behaviors.

## 5.4. Amplification of discovered minority attributes during generation

*Table 3.* Amplification on WinoBias via RAIGen-guided prompt revision. The top table reports results for SD v1.4 and the bottom table reports results for SDXL.

| Prompt used | NLL (↑) | Dev. ratio (↓) | Align. (↑) |
|---|---|---|---|
| Base prompt | 1.917 | 0.50 | **20.30** |
| Ours-Revised | **1.935** | **0.22** | 19.80 |

| Prompt used | NLL (↑) | Dev. ratio (↓) | Align. (↑) |
|---|---|---|---|
| Base prompts | 1.812 | 0.49 | 27.26 |
| Ours-Revised | **1.852** | **0.23** | 26.89 |

We investigate whether minority attributes identified by RAIGen can be used to amplify underrepresented modes in the generative distribution. Using RAIGen's language annotations, we perform lightweight prompt revision guided by Llama 4-Scout, which injects minority descriptors into the input text. While RAIGen is agnostic to the downstream mitigation strategy (Chuang et al., 2023; Friedrich et al., 2023), prompt revision provides a simple test of whether reintroducing minority concepts changes the model's generations in the intended direction. We evaluate on COCO and WinoBias by uniformly sampling images from (i) the original prompt and (ii) the revised prompt. For each setting, we report: (i) negative log-likelihood scored under the base prompt, (ii) deviation from a uniform attribute distribution, and (iii) CLIP alignment to the original prompt (see Appendix F.2 for metric details).

Table 3 shows that RAIGen-guided prompt revision substantially reduces attribute-presence deviation for both SD v1.4 and SDXL, indicating that the revised prompts increase coverage of minority attributes and move generations closer to a balanced attribute distribution. RAIGen-guided generations have higher negative log-likelihood under the base prompt, indicating that the amplified samples occupy lower-density regions of the original distribution and a small drop in prompt alignment, suggesting that we can meaningfully surface rare modes while largely preserving the original prompt semantics. Additional results on COCO are provided in Appendix G.4. Overall, RAIGen provides a practical mechanism for amplifying underrepresented attributes beyond predefined fairness categories.

## 5.5. Applicability of RAIGen to Transformer-based Diffusion Models

We further evaluate RAIGen on a transformer-based diffusion backbone by applying it to FLUX.1-schnell, whose denoiser follows a DiT-style transformer architecture. For a fixed prompt $c$ (e.g., "a photo of a doctor"), we run the model with 4 denoising steps, cache intermediate hidden states from `transformer.transformer_blocks.18` following Surkov et al. (2025), and train an MSAE on these activations. We then rank coarse-level features using the RAIGen minority score. On the prompt "a photo of a doctor", the discovered attributes achieve Attribute Presence = 0.11, indicating that RAIGen isolates concepts that are encoded in FLUX representations yet rarely expressed under default sampling. As shown in our qualitative results (Fig. 5), the top-ranked neurons correspond to coherent underrepresented modes visible in both top-activating images and attribution maps, including *female doctors with curly/afro-textured hair* and *masked doctors*.

Interestingly, we observe a higher fraction of high-scoring but weakly interpretable neurons in FLUX compared to U-Net–based diffusion models: several high-scoring candidates exhibit diffuse or non-localized heatmaps in the qualitative analysis, making semantic verification less reliable. We attribute this, in part, to architectural differences in representational structure. U-Nets expose an explicit spatial bottleneck whose channels are naturally aligned with localized image regions, making feature attribution and heatmap visualization comparatively well-behaved. In contrast, the chosen FLUX transformer hook point may encode features that are less spatially grounded than U-Net bottlenecks, especially under few-step sampling. This suggests that rare-attribute discovery in transformer diffusion may benefit from more systematic selection of hook points across blocks and submodules (e.g., attention vs. MLP streams), which we leave for future work. Overall, these results indicate that RAIGen generalizes beyond U-Net architectures and provides a viable approach for rare-attribute auditing in transformer-based diffusion models.

## 6. Conclusions

We propose RAIGen, a framework for minority attribute discovery in diffusion models that combines Matryoshka Sparse Autoencoders with a novel minority score to identify features encoded in latent representations but underrepresented during generation. Unlike prior work focused on fixed fairness categories or majority trends, RAIGen uncovers rare, semantically meaningful attributes directly from internal activations. Through quantitative and qualitative analyses, including a user study showing that RAIGen discovers user-aligned attributes that are underexpressed in generations, we demonstrate that RAIGen reveals fairness, stylistic, and cultural minorities across Stable Diffusion variants, and facilitates targeted amplification via prompt revision. By grounding discovery in model representations, RAIGen complements LLM-based bias tools and lays the foundation for hybrid auditing frameworks that bridge external priors with internal model dynamics.

## Impact Statement

This paper introduces RAIGen, a representation-based method for identifying and amplifying underrepresented attributes encoded in text-to-image diffusion models via interpretable neuron representations. The primary intended positive impact is to provide a general tool for uncovering suppressed or rare semantic factors in generative models, improving transparency about what models represent and what they tend to neglect.

RAIGen can only surface attributes that are already encoded in a model's internal representations. As a result, socially salient minorities that are not learned by the model may not be discovered. This motivates combining representation-grounded discovery with complementary approaches that incorporate external semantic priors, such as LLM-based tools, to obtain a broader view of underrepresentation.

The same capabilities could be misused. Amplifying rare attributes may enable the deliberate production of sensitive, stereotyped, or culturally harmful imagery, or facilitate targeted generation of protected traits in contexts where this is undesirable. More generally, feature-level control could be used to steer models toward propaganda-style messaging or other harmful content. Overall, we view RAIGen as a representational analysis and control tool whose societal impact will depend on the safeguards, governance, and deployment practices surrounding generative models.

## Acknowledgements

Serge Belongie and Silpa Vadakkeeveetil Sreelatha are supported in part by the Pioneer Centre for AI, DNRF grant number P1. Silpa Vadakkeeveetil Sreelatha also thanks the ELLIS PhD Program for support and acknowledges travel support from the ELIAS Mobility Fund and the Turing Mobility Scheme (2024/25) from the UK. The authors acknowledge the use of resources provided by the Isambard-AI National AI Research Resource (AIRR) (McIntosh-Smith et al., 2024). Isambard-AI is operated by the University of Bristol and is funded by the UK Government's Department for Science, Innovation and Technology (DSIT) via UK Research and Innovation; and the Science and Technology Facilities Council [ST/AIRR/I-A-I/1023]. We also thank Stella Frank for proofreading the paper and providing valuable suggestions.

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

# A. Appendix

In the primary text of our submission, we introduce RAIGen, a framework for uncovering minority attributes encoded in the internal representations of diffusion models. To preserve clarity and conciseness in the main paper, we provide an extensive appendix that complements the core manuscript. The appendix includes additional experiments, detailed implementation protocols, broader qualitative analyses, and deeper ablations that could not fit within the page limits. Together, these materials extend the discussion in the main text by offering a fuller view of our methodology, empirical validation, and implications.

# B. Discussion

**Meaningful discoveries vs. generic long-tail behavior** A natural question is whether RAIGen surfaces socially or semantically meaningful rare attributes, or merely identifies low-frequency visual patterns that reflect generic long-tailed generation. We argue that, in the generative setting, this distinction cannot be drawn a priori. Unlike discriminative long-tail recognition, where rare classes are identified by counting labels, generation has no predefined categories over which to measure imbalance: a "class" in the generative tail may be a viewpoint, a species, a hair texture, or a contextual cue, and is itself unknown before discovery. Several RAIGen findings are not predictable from any obvious semantic prior. For the prompt "a photo of an assistant" (Fig. 12, SDXL), RAIGen surfaces *animals* (cats, dogs) as a suppressed mode, a learned association no semantic prior would anticipate. For "a small yellow bird sitting on a tree branch" (Fig. 13), it reveals species collapse onto a narrow visual prototype. These discoveries would be difficult to anticipate through manual specification of fairness categories. Whether a given suppressed attribute warrants intervention is inherently application-dependent: suppressed species diversity matters for an educational platform but not for a recruitment tool; suppressed hair textures matter for stock photography but not for autonomous driving. RAIGen's role is to surface the full inventory of suppressed modes, and the practitioner provides the domain-specific judgment over which entries matter.

**Practical utility of RAIGen** Beyond cross-model auditing (Sec. G.2 of Appendix) and amplification (Sec. 5.4), RAIGen serves a broader role as an upstream discovery layer for any pipeline that needs to know which attributes are encoded but suppressed in a target model. **1. Identifying which attributes require specification.** Every unspecified dimension in a prompt is a dimension where bias can operate silently. This is the setting in which all open-set bias auditing operates (D'Incà et al., 2024): prompting "a female doctor" resolves gender, but skin tone, hair texture, clothing, and setting remain unspecified, and each subsequent specification opens new unspecified dimensions. The space of possible suppressions is too large for manual inspection. Without a systematic discovery tool, practitioners have no principled way to know which dimensions are biased and which are not. RAIGen surfaces these dimensions explicitly, and its discoveries persist for reuse in amplification, cross-model tracking, or targeted auditing of specific deployment prompts (e.g., "doctor" or "CEO" for recruitment imagery). **2. Targeted rare generation:** RAIGen's discoveries directly address a gap in existing rare-generation pipelines. Methods such as Minority Guidance (Um et al., 2024) amplify low-density regions but do not identify which attributes are rare or worth amplifying. We validate this experimentally in Appendix G.5: for "a photo of an assistant," Minority Guidance amplifies "animals" due to statistical rarity (attribute presence 0.382 vs. 0.150 for the SDXL baseline), but this attribute is contextually irrelevant for a profession task. Methods such as Rare-to-Frequent (R2F) (Park et al., 2025) and ToRA (Kang et al., 2025) produce high-fidelity rare generations but require prior knowledge of which concepts to target as input. RAIGen complements both lines of work: rather than replacing them, it provides the upstream discovery step they assume, automating the manual specification of what to generate. This positions RAIGen as a diagnostic and auditing tool whose outputs can serve as inputs to dedicated generation methods.

# C. Future work

A promising future direction is the integration of LLM-based and representation-based approaches for minority attribute discovery. While RAIGen surfaces attributes that are internally encoded but suppressed during generation, LLM-based tools offer a complementary perspective by drawing on external world knowledge to hypothesize socially salient or semantically expected minorities.

Since no existing method explicitly targets minority attribute discovery via LLMs, we perform a preliminary study by adapting OpenBias, originally designed for majority (overrepresented) feature identification for this purpose. While not validated for minority detection, we repurpose its language-based pipeline to surface expected but underrepresented attributes (e.g., non-binary cashiers, Hispanic teachers) and compare them against those discovered by RAIGen (e.g., doctors in

vintage attire, teachers outdoors). Adapting OpenBias for minority discovery yields attributes with near-zero presence in generated samples (0.021 for WinoBias, 0.058 for COCO), reflecting concepts that the model fails to encode altogether. This highlights the complementarity of the two approaches: LLM-based methods surface expectation-driven minorities aligned with human priors, while RAIGen reveals underexpressed concepts that are internally encoded. Together, they provide a broader view of underrepresentation, motivating future work on unified frameworks that combine external semantic priors with internal representation-grounded discovery.

Our current pipeline incurs additional compute because minority discovery is prompt-specific, requiring training a separate sparse autoencoder (SAE) for each auditing setup. Future work could reduce this overhead by amortizing representation learning across prompts, for example, training a shared SAE on a diverse prompt mixture and using lightweight prompt-conditioned latent selection.

## D. Limitations

While RAIGen provides a systematic framework for discovering minority attributes in diffusion models, it is not without limitations. First, our approach only captures attributes that are encoded in the model's internal representations, meaning that socially salient fairness categories entirely absent from the representation space (e.g., non-binary identities, certain cultural groups) will not be surfaced. This limits the scope of fairness auditing to what the model has already learned, rather than what society may expect. Second, since RAIGen relies on sparse autoencoders for interpretability, the quality and granularity of discovered attributes depend heavily on what the autoencoder itself captures. Attributes may be fragmented, merged, or overlooked depending on the sparsity budget and training dynamics, and alternative architectures could yield different results. Third, even among the minority neurons that are identified, some may remain noisy or ambiguous despite our filtering and validation steps. In the absence of ground truth annotations for minority attributes, it is difficult to rigorously quantify coverage and precision of the discovered neurons. These limitations should be considered when applying RAIGen, particularly in fairness auditing and downstream interventions. Nonetheless, they do not diminish the standalone utility of the approach in surfacing underrepresented concepts, but rather highlight opportunities to further strengthen it when combined with complementary tools.

## E. Experimental details

This section provides a detailed account of the experimental setup, including datasets, training procedures, and hyperparameter choices, to ensure that our results can be reliably reproduced.

### E.1. Discovery of minority attributes

We utilize 10 WinoBias professions and 50 COCO prompts to investigate the effectiveness of RAIGen. For WinoBias, prompts are constructed in the form "A photo of a *profession*", where the profession is drawn from the benchmark, following prior text-to-image generation frameworks. For COCO, we use the original prompt captions without modification. For each prompt in WinoBias and COCO, we generate 5000 images together with their bottleneck representations of size $1280 \times 8 \times 8$ across all timesteps. Each spatial location at each timestep is treated as an independent sample, yielding a 1280-dimensional vector that serves as input to the MSAE. This setup is adopted by (Cywiński & Deja, 2025). We train MSAE using a sparse latent feature space of size $1280 \times 16$, employing two sparsity levels: a coarse level with $k_1 = 2048$ neurons, and the remaining neurons allocated to the fine level. We adopt the official implementation provided by (Bussmann et al., 2025) for training, with the following hyperparameters: 5 training epochs, an effective batch size of $4,096$, a learning rate of $3 \times 10^{-4}$, and an auxiliary penalty weight of $\frac{1}{32}$.

Following training, we generate $5,000$ images along with their bottleneck representations of size $1280 \times 8 \times 8$ at timestep=49. The number of timesteps during sampling from diffusion models is set to 50. To compute semantic distinctiveness, we use the CLIP ViT-L/14 model to extract embeddings for the top-activating images of each neuron. For interpretability, we use GPT-5.2 to annotate the minority neurons. During the pruning, we fix a cosine distance threshold of 0.003. To further ensure semantic relevance, we restrict the set to neurons whose minority scores exceed the 90th percentile. Hyperparameters were selected through sweeps and guided by heuristics, and validated using interpretability checks.

We now provide the prompt that we used to annotate our neurons using GPT-5.2 :

```
You are a JSON-only generator. You are not allowed to explain anything, write markdown, or
    comment. Only return a single valid JSON object. You have to analyze a specific
```

```
    neuron from a sparse autoencoder trained on profession-related generated images. Each
    neuron activates in response to specific, consistent visual features that appear
    across its top-activating images, as confirmed by corresponding heatmaps.

- You are provided with:
  - Top-activating images (first row): strongest activations for this neuron
  - Heatmaps (second row): regions most responsible for the neuron activation

- Your job is:
  1. To carefully observe the top-activating images and heatmaps, and identify visually
     consistent attributes that correlate with the neuron activation.
  2. To generate a modified version of the base prompt that includes these attributes
     naturally and precisely.
  3. To output a flat list of non-redundant keywords capturing only the consistent
     attributes.

- Strict requirements:

  - The identified attributes must be:
    - Clearly and consistently visible across the top-activating images
    - Highlighted or partially supported by the heatmap attention
    - Not already implied by the base prompt
    - Not a core object/tool expected for the profession

{{
    "neuron_id": "{neuron_id}",
    "input prompt": "{base_prompt}",
    "identified_attribute": "<short, precise description of all consistent visual
        attributes across the images>",
    "suggested_prompt": "<the base prompt modified to include these attributes naturally
        >",
    "keywords": ["<keyword_1>", "<keyword_2>", "..."]
}}

### Example Outputs

Input Prompt: "a photo of a doctor"

Example 1:
{{
    "neuron_id": "2041",
    "input prompt": "a photo of a doctor",
    "identified_attribute": "female doctor",
    "suggested_prompt": "a photo of a female doctor",
    "keywords": ["female"]
}}
```

## F. Evaluation details

In this section, we provide additional evaluation details that we utilized to investigate the effectiveness of RAIGen in identifying underrepresented attributes.

### F.1. Discovery of Minority attributes

We evaluate the effectiveness of RAIGen at discovering minority attributes using Attribute Presence, which measures how often the discovered attributes are expressed in generated samples. We use the minority-attribute annotations produced by GPT-5.2 (Section 4 of the main paper) to form candidate attribute queries. For each generated image and candidate attribute, we perform VQA-style verification by providing the image and the attribute to a vision-language evaluator and asking: *"Is the attribute present in the image?"* We record the resulting binary prediction and aggregate over images to obtain an attribute-wise presence score, then aggregate across attributes for a given prompt, and finally average across prompts to report the overall Attribute Presence. Lower values correspond to attributes that are more underrepresented under default

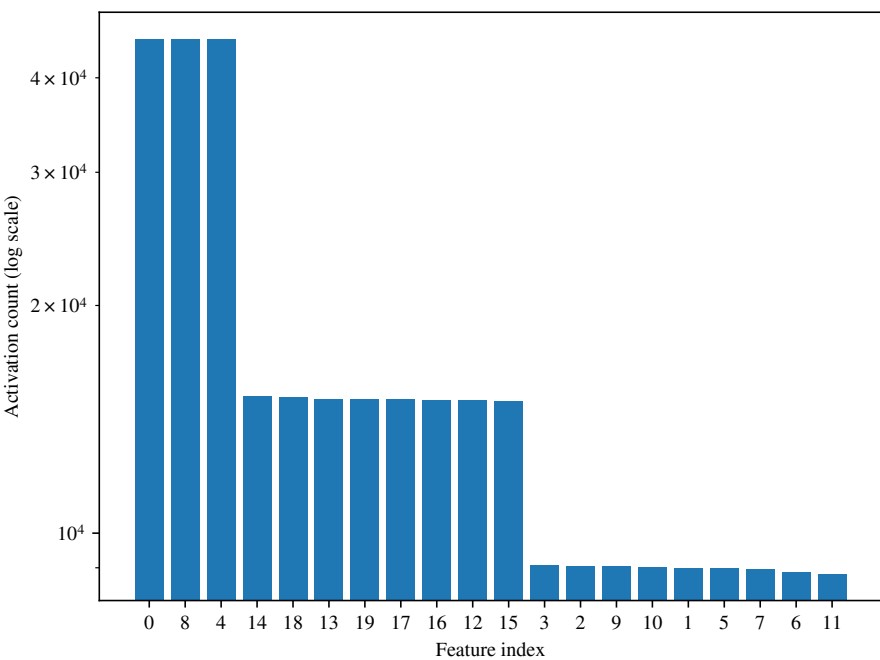

*Figure 6.* **Long-tailed feature frequencies in the toy setting.** Activation counts for each ground-truth feature over $N{=}300{,}000$ samples, sorted in descending order.

sampling.

To reduce evaluator-specific biases, we use an ensemble of vision-language models rather than relying on a single VQA system. Concretely, we query both Llama 4-Scout and Qwen3-VL-8B-Instruct for each image–attribute pair and combine their outputs via majority vote.

### F.2. Amplification of Minority attributes

We evaluate the effectiveness of amplifying the minority attributes discovered by RAIGen during generation using three metrics: likelihood, discrepancy from a uniform distribution, and CLIP alignment. In the amplification setting, we generate images for each attribute under two conditions: (i) the base prompt, and (ii) a revised prompt augmented with the discovered minority attribute. For each image, one of these two conditions is randomly selected, ensuring a balanced mixture of base and revised generations across the evaluation set.

**Negative log-likelihood.** For each neuron identified as encoding a minority attribute, we select its top-10 activating images and compute their exact log-likelihoods conditioned on the prompt using the PF-ODE estimator of Song et al. (2021). The likelihoods are aggregated across images for each prompt. For each attribute, we compute the average log-likelihood across all generated images and then aggregate across all attributes under consideration. We report the negative log-likelihood in bits per dimension. Higher values indicate that the amplified samples remain in low-density regions of the distribution, consistent with their underrepresented nature.

**Discrepancy.** Discrepancy measures how evenly images are distributed between the base and revised prompts. Intuitively, if amplification is successful, the distribution of samples across attributes should approach uniformity. Formally, this metric is defined analogously to fairness discrepancy measures in prior works (Parihar et al., 2024), where smaller values indicate better balance between base and revised generations.

**CLIP Alignment.** Finally, we measure the semantic alignment between the generated images and their corresponding textual prompts using CLIP embeddings. Higher alignment scores indicate that amplification strengthens the consistency between the intended minority attribute and the visual output, without sacrificing prompt fidelity.

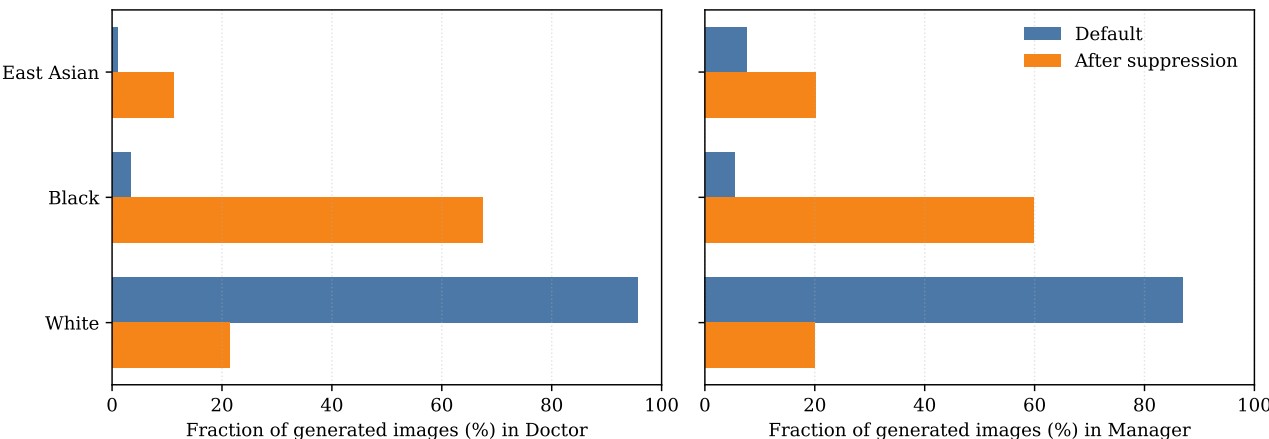

*Figure 7.* Attribute prevalence under default sampling vs. after naive suppression of the dominant attribute (White) for SDXL generations for Doctor and Manager. Suppressing the majority attribute reduces White prevalence substantially, but the resulting increase is concentrated in some minorities (notably Black) rather than uniformly amplifying all minorities (e.g., East Asian).

For both WinoBias and COCO prompts, we evaluate amplification using the top-5 minority attributes per prompt, generating 100 images for each prompt under this setting.

## G. Additional experiments

In this section, we provide additional experiments that we performed to investigate the effectiveness of RAIGen in identifying underrepresented attributes. The experiments are performed on the representations extracted using SD v1.4.

### G.1. Majority-suppression does not uniformly amplify minority attributes

We illustrate a limitation of majority-attribute suppression using a controlled study on SDXL. For each occupation prompt (*"a photo of a doctor"* and *"a photo of a manager"*), we generate 500 images with default sampling and estimate the prevalence of demographic attributes (White, Black, East Asian) using an external attribute annotator. We then apply a naive prompt-based suppression intended to reduce the dominant attribute (White): concretely, we add the negative prompt ``white-race, caucasian, european, blonde`` and re-sample 500 images under the same settings, which reduces the frequency of White-presenting subjects.

The results are reported in Figure 7. As expected, default sampling is highly skewed toward White subjects (Doctor: 95.6%, Manager: 87.0%), consistent with prior observations (AlDahoul et al., 2025). After naive suppression of the dominant attribute, the prevalence of White subjects drops substantially, but the freed probability mass does not translate into a uniform increase across minority groups. Instead, it shifts disproportionately toward a subset of minorities (e.g., a sharp increase in Black subjects) while others remain comparatively underrepresented (e.g., only a modest increase in East Asian subjects). This pattern is the generative-model analogue of the Whac-A-Mole dilemma identified in discriminative settings by Li et al. (2023); Kappiyath et al. (2025): mitigating one dominant factor does not eliminate the broader imbalance, but redistributes it onto a different sub-attribute. Single-target suppression therefore does not reliably surface or amplify all minority attributes that may be encoded in the model, motivating the need for systematic minority-attribute discovery before any intervention is applied.

### G.2. Cross-Model Analysis of Minority Attributes

In this section, we use RAIGen as a systematic auditing framework to compare minority attribute representations across diffusion architectures. Focusing on the profession of *Doctor*, we apply RAIGen independently to SD v1.4, v2.1, and XL to identify minority neurons and annotate their semantics. We then collect the unique set of discovered attribute annotations across all models to form a unified attribute vocabulary. For each attribute in this set, we measure its presence in each model by generating 1000 samples from the corresponding model to evaluate attribute presence. For every candidate attribute in the unified vocabulary, the Llama 4-Scout model is provided with the generated image and asked whether the attribute is

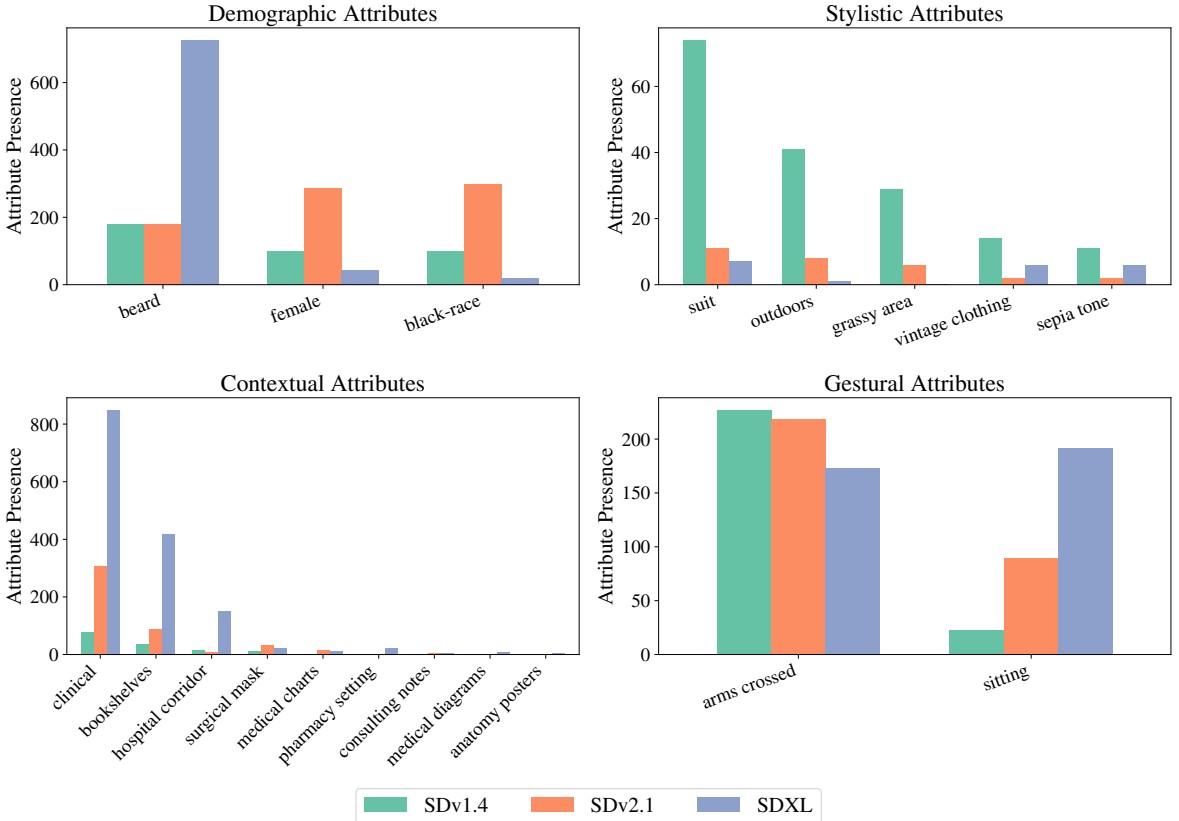

*Figure 8.* Category-level presence of minority attributes in images of doctors across SD versions. Demographic, stylistic, contextual, and gestural attributes reveal distinct representational shifts.

present using the prompt *"Is the attribute present in the image?"*, and we record whether the attribute is detected. We repeat this process across all attributes. This produces an occurrence count that allows us to compare how frequently each attribute manifests across models.

The results are summarized in Figure 8. Demographic attributes show the strongest change: SD v1.4 heavily underrepresents female and black doctors, partially corrected in SD v2.1, while SDXL introduces new imbalances, such as overemphasis on traits like beards. Stylistic minorities follow the reverse trend—attributes like sepia tone, vintage clothing, and outdoor settings appear in SD v1.4 but are nearly absent in SDXL, which favors standardized, formal portrayals. Contextual minorities shift most dramatically: diverse backdrops (e.g., bookshelves, hospital corridors) in early models are replaced by a dominant clinical office setting in SDXL, indicating contextual homogenization. In contrast, gestural attributes such as arms crossed or sitting remain stable across versions, suggesting pose features are robustly preserved. By surfacing minority attributes and enabling their tracking across model families, RAIGen reveals how architectural changes and scaling can shift underrepresentation rather than resolve it. Gains in demographic balance may trade off with stylistic or contextual diversity, highlighting the need for auditing of all forms of underrepresentation beyond narrow fairness categories.

### G.3. Why Low Minority Scores Do Not Capture Majority Features

While the *Minority Score* is well suited to isolate underrepresented features (low frequency, high distinctiveness), we investigate whether neurons with the lowest scores might instead recover dominant attributes. To this end, we utilize RAIGen to identify minority neurons from the intermediate representations of diffusion models for the prompt *"A photo of a Doctor"*. We then examine the bottom-ranked neurons from our trained MSAE at coarse sparsity levels, where semantic content is most fully expressed. We visualized their activations and corresponding heatmaps, using Figure 9 to illustrate representative cases.

Empirically, we observe that the dominant features appear duplicated, fragmented, or spread across many neurons whereas

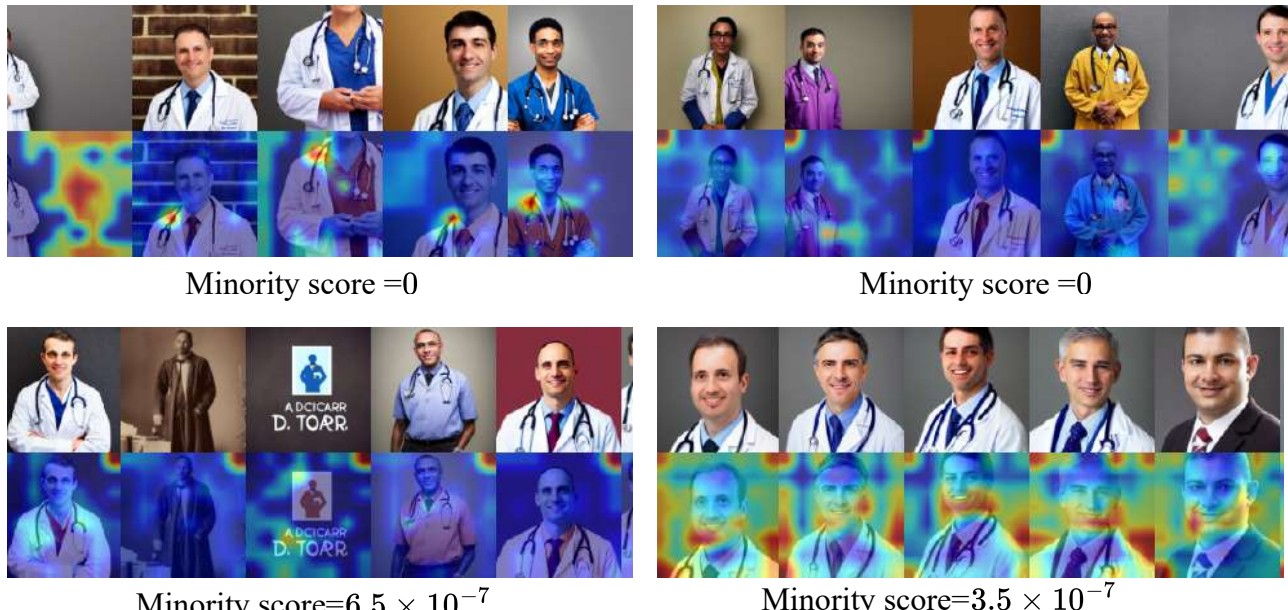

Minority score =0       Minority score =0

Minority score=$6.5 \times 10^{-7}$       Minority score=$3.5 \times 10^{-7}$

*Figure 9.* Low-scored (majority) neurons identified by RAIGen for the prompt *"A photo of a doctor"*. Unlike minority neurons, their activations and heatmaps are diffuse, fragmented, and fail to capture coherent semantic attributes.

minority features emerge as sharp, semantically coherent units. This arises because the majority attributes correspond to broad, high-variance regions of the diffusion representation space, which are distributed across multiple correlated directions rather than concentrated along a single axis. As a result, the neurons with the lowest Minority Scores remain diffuse and unreliable for interpretation, often localizing to narrow, fine-grained patches in the heatmaps rather than capturing the attribute holistically. This variation persists even under strong sparsity, forcing the model to distribute reconstruction responsibility across several correlated neurons. By contrast, minority concepts occupy compact, lower-variance regions with little redundancy, enabling the MSAE to assign a single neuron to capture the full feature without incurring large reconstruction penalties. Consequently, low scores do not provide clean access to dominant ones and hence, the framework is expressly tailored for the discovery of minority attributes, and not for the recovery of interpretable majority features.

*Table 4.* Amplification on COCO via RAIGen-guided prompt revision. The top table reports results for SD v1.4 and the bottom table reports results for SDXL.

| Prompt used | NLL ($\uparrow$) | Dev. ratio ($\downarrow$) | Align. ($\uparrow$) |
|---|---|---|---|
| Base prompt | 1.943 | 0.53 | 26.93 |
| Ours-revised | 1.962 | **0.25** | 25.96 |

| Prompt used | NLL ($\uparrow$) | Dev. ratio ($\downarrow$) | Align. ($\uparrow$) |
|---|---|---|---|
| Base prompt | 1.819 | 0.51 | 20.52 |
| Ours-revised | 1.864 | **0.28** | 19.83 |

### G.4. Additional results on amplification of minority attributes on COCO

To complement our evaluation in Section 5.4 of the main paper, we assess how the minority attributes discovered by RAIGen can be utilized to amplify their generation. We consider the top-5 minority attributes identified by RAIGen for 50 COCO prompts. For each attribute, we generate images with and without revised prompts uniformly and compute the likelihood under the base prompt distribution. We also compute attribute deviation from uniformity and CLIP alignment. As in the main paper, likelihood values are compared against a baseline of 500 randomly sampled images for each COCO prompt. The results are summarized in Table 4.

Table 4 shows that RAIGen-guided prompt revision substantially reduces attribute-presence deviation for both SD v1.4

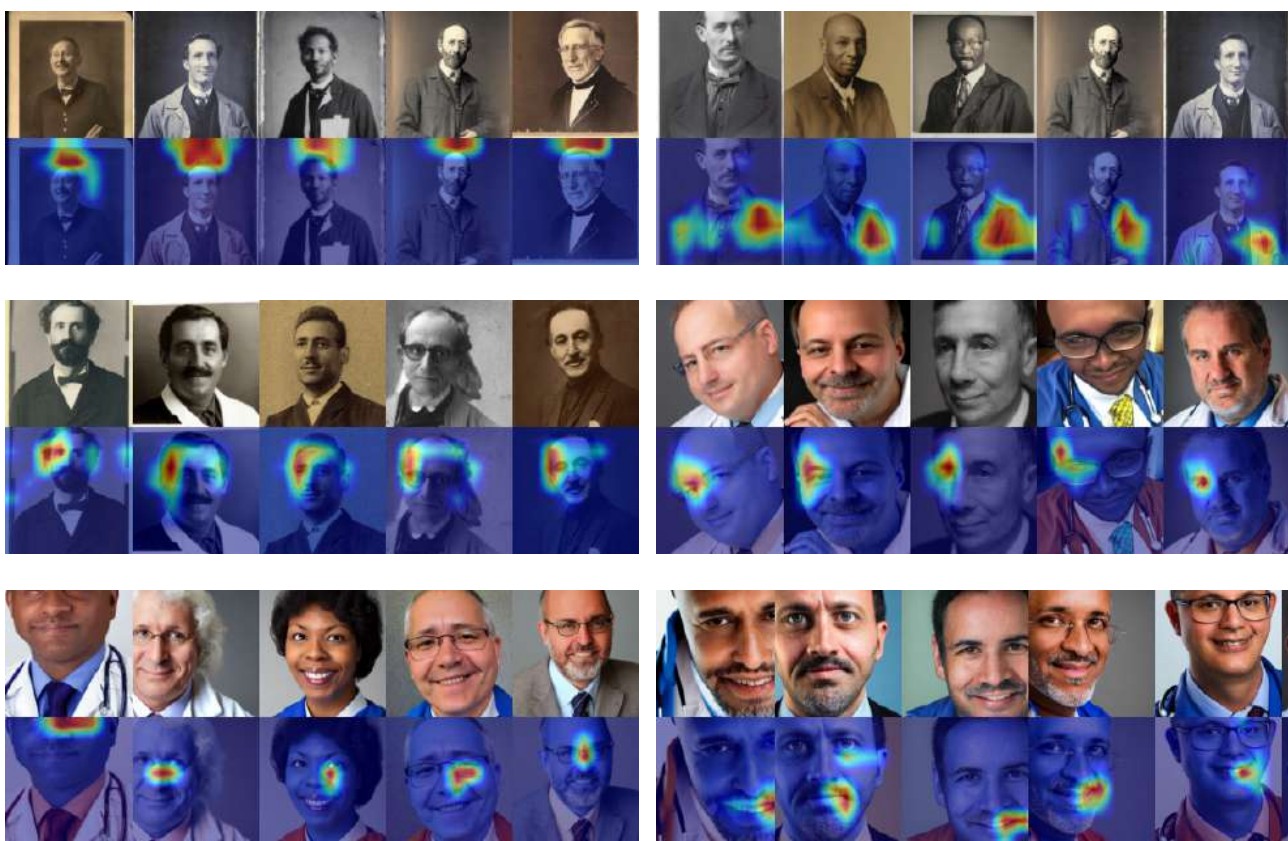

*Figure 10.* Visualizations of top-activating samples and corresponding heatmaps for top minority neurons identified using a standard SAE for RAIGen. Each pair shows the top-activating images and the neuron's activations. While the neurons capture meaningful localized features (e.g., facial details or specific contextual elements), they frequently fragment broader concepts across multiple neurons, illustrating reduced interpretability compared to MSAE.

and SDXL, indicating improved coverage of minority attributes and generations closer to a balanced attribute distribution. At the same time, prompt revision using RAIGen discovered atributes increases NLL when scored under the base prompt for both models, consistent with shifting samples toward lower-density regions of the original distribution (i.e., surfacing underrepresented modes). Prompt alignment decreases modestly, which is expected because alignment is measured against the original prompt while our revised prompts explicitly inject additional minority descriptors. As in the main paper, our goal is not to optimize the mitigation strategy but to demonstrate that RAIGen's discovered attributes can be used to reliably amplify rare modes in an open-domain setting; more targeted interventions (e.g., controlled decoding or representation steering) could further preserve prompt fidelity while amplifying minority attributes.

### G.5. Comparison with statistical rarity sampling: Minority Guidance

Sampling-based methods such as Minority Guidance (Um et al., 2024) amplify generations toward low-density regions of the conditional distribution using statistical signals (e.g., classifier-free guidance scaled by an inverse-likelihood term). A natural question is whether such methods make a discovery framework like RAIGen redundant: if low-density samples are already accessible via sampling alone, why train an MSAE and rank neurons?

We argue that Minority Guidance and RAIGen solve different problems. Minority Guidance amplifies any attribute satisfying its statistical-rarity heuristic, regardless of whether that attribute is contextually meaningful or socially relevant. RAIGen instead identifies which attributes are encoded but suppressed, leaving the choice of which to amplify to the practitioner. We validate this empirically on the prompt "a photo of an assistant" (SDXL), for which RAIGen surfaces *animals* (cats, dogs) as a learned suppressed mode (see Fig. 12, "assistant"). We then generate samples from SDXL with Minority Guidance applied at the same prompt and compute Attribute Presence for "animals."

*Table 5.* Attribute Presence for "animals" on the prompt "a photo of an assistant" (SDXL). Minority Guidance amplifies "animals" to 0.382, but "animals" is contextually irrelevant for a profession task. RAIGen's role is to surface the attribute (so practitioners can decide whether to amplify it), not to amplify all statistically rare attributes indiscriminately.

| Method | Attribute Presence ("animals") |
|---|---|
| Base SDXL | 0.150 |
| Minority Guidance | 0.382 |

As shown in Table 5, Minority Guidance more than doubles the presence of "animals" over the base SDXL distribution. Without prior knowledge that "animals" is the attribute being amplified, a practitioner using Minority Guidance to audit or diversify "assistant" generations would obtain outputs more frequently containing animals, an outcome that is statistically warranted but contextually problematic for an occupation-related task. RAIGen's contribution here is upstream: it identifies that this association exists in the model, so a practitioner can decide whether to amplify or suppress it. The two approaches are therefore complementary rather than competing: RAIGen identifies what is suppressed, and sampling-based methods like Minority Guidance can then amplify attributes once they are chosen.

### G.6. Comparison with image-level clustering

To test whether neuron-level decomposition is necessary for rare-attribute discovery, we compare RAIGen against a simpler image-level baseline: $k$-means clustering on CLIP image embeddings of generated samples. For each prompt, we generate 5,000 images on SDXL, compute their CLIP-ViT-L/14 embeddings, and run $k$-means with $k = 50$. We treat the ten smallest clusters as minority-attribute candidates (the image-level analogue of selecting rarely activated neurons), annotate each cluster using the same protocol as for RAIGen neurons (Sec. 4), and compute Attribute Presence on the resulting annotations.

*Table 6.* Mean Attribute Presence ($\downarrow$) of minority attributes discovered by CLIP $k$-means vs. RAIGen across six WinoBias professions on SDXL. RAIGen consistently isolates more strongly suppressed attributes.

| Profession | CLIP $k$-means | RAIGen |
|---|---|---|
| Doctor | 0.30 | 0.15 |
| Sheriff | 0.50 | 0.18 |
| Farmer | 0.69 | 0.25 |
| Attendant | 0.47 | 0.13 |
| Nurse | 0.32 | 0.23 |
| Assistant | 0.35 | 0.15 |
| Average | 0.37 | 0.18 |

Table 6 shows that RAIGen achieves lower Attribute Presence than CLIP $k$-means across all six professions, with an average gap of 0.19 in absolute presence. Qualitatively, image-level clusters frequently mix dominant and minority instances within a single cluster, since CLIP embeddings cluster by overall visual similarity rather than by individual attribute factors. For example, a cluster surfaced for "doctor" often contains a mix of attire, demographics, and contexts, making it difficult to isolate which attribute makes that cluster "rare." Several small clusters are also uninterpretable at the attribute level where they correspond to low-frequency visual patterns (e.g., specific lighting conditions, framing artifacts) without a coherent semantic attribute. By contrast, RAIGen's MSAE-based decomposition factorizes generations along axes that align more cleanly with individual semantic concepts, yielding minority neurons that correspond to coherent, interpretable attributes. These results confirm that neuron-level decomposition is necessary for systematic minority-attribute discovery and image-level clustering, while simpler, does not surface the same set of suppressed attributes.

### G.7. RAIGen using vanilla SAE

While our primary experiments utilize MSAEs for hierarchical semantic decomposition, we also conduct a parallel analysis using standard SAEs to evaluate how the choice of decomposition method affects the discovery of minority attributes. We apply RAIGen to intermediate representations of diffusion models for the prompt *"A photo of a Doctor"*. In this setting, rather than using MSAE, we employ a vanilla SAE to identify neurons associated with underrepresented attributes. We visualize several of the top minority neurons discovered by our approach, with the results summarized in Figure 10.

To evaluate the role of the decomposition method, we repeated our analysis using a standard Sparse Autoencoder (SAE) in place of the Matryoshka SAE. As shown in Figure 10, the SAE was still able to surface minority attributes under our *Minority Score*, with neurons capturing features such as eyeglasses and contextual backdrops. However, consistent with observations in (Bussmann et al., 2025), SAE representations were considerably more fragmented, often distributing a single concept across multiple neurons. For example, we identified cases where one neuron responded primarily to the right eye and another to the left eye; although both were flagged as minority neurons due to their selective activations, they did not reflect genuine semantic minorities but rather over-fragmented features. This fragmentation reduced semantic coherence, making annotation less straightforward and limiting interpretability. Overall, these results demonstrate that while SAEs can uncover minority attributes, their tendency to fragment concepts across multiple neurons can be misleading, as neurons may appear to represent minorities while in fact capturing only partial or redundant features. In contrast, MSAEs mitigate this issue by providing hierarchical control over granularity and emphasizing global semantic features, enabling more faithful and interpretable discovery of genuinely underrepresented attributes.

### G.8. Robustness to choice of vision-language encoder

The semantic distinctiveness term in the Minority Score (Eq. 3) is computed against CLIP image embeddings. To test whether RAIGen's discoveries depend on the specific embedding model, we replace CLIP-ViT-L/14 with SigLIP (Zhai et al., 2023) and re-run the full discovery pipeline on the prompt "a photo of a doctor" (SDXL), keeping all other components (MSAE, activation frequency, ranking, redundancy filtering) unchanged. We then compare the top-10 minority neurons surfaced by each variant.

We observe that 9 of the top-10 neurons are recovered by both encoders, and the single non-overlapping neuron is non-interpretable under both. We attribute this stability to the nature of the distinctiveness term itself. It measures a coarse semantic contrast between a neuron's activation-weighted centroid and the dataset-wide centroid, a signal that any vision-language model trained on large-scale image–text pairs captures reliably. The agreement between two encoders trained with different contrastive objectives (CLIP's symmetric InfoNCE vs. SigLIP's sigmoid loss) indicates that RAIGen's discoveries reflect stable structure in the MSAE feature space rather than embedding-specific artifacts. Exploring more principled distinctiveness formulations (e.g., information-theoretic or contrastive-objective-aware) remains a promising direction for future work.

### G.9. Ablation on MSAE coarse level size for RAIGen

In this section, we perform an ablation on the number of neurons in the coarse sparsity level in MSAE which we utilize for RAIGen to understand its effect on the minority attribute discovery. We ablate the MSAE coarse sparsity level size $k_1 \in \{1280, 2048, 10240\}$ while fixing the sparse feature dimension to 20480 (input representation size = 1280, expansion factor = 16), and evaluate on the prompt *"A photo of a doctor"*. As shown in Table 7, attribute presence increases at larger $k_1$. Qualitatively, we observed that some demographic attributes (e.g., Black skin tone) are missed at $k_1$=1280 but captured at $k_1$=2048. Balancing coverage against fragmentation, we adopt $k_1$=2048 as the default, which reliably recovers key demographics without the over-fragmentation seen at very large $k_1$.

| Number of neurons in $k_1$ | Attribute presence |
|---|---|
| 1280 | 0.15 |
| 2048 | 0.15 |
| 10240 | 0.20 |

*Table 7.* Ablation on the number of neurons in coarse sparsity level $k_1$ in MSAE for RAIGen. Attribute presence is computed as in our evaluation protocol.

### G.10. Ablation on different components of the Minority Score

This section evaluates the design choice behind our *Minority Score*, which combines activation frequency and semantic distinctiveness to identify minority-associated neurons. Our goal is to recover neurons that correspond to underrepresented attributes while remaining semantically meaningful. We compare three neuron-ranking strategies under the prompt *"A photo of a doctor"* in SDXL: frequency-only ranking using $\nu_i$, semantic-distinctiveness based ranking using $d_i$, and the combined Minority Score $s(z_i) = d_i \cdot (1 - \nu_i)$ and consider top 10 neurons discovered in each case. We evaluate each selection using (i) the fraction of neurons labeled `No identified attribute` by the annotation model which we

| Method | Uninterpretable ↓ | Interpretable ↑ | Mean attribute presence ↓ |
|---|---|---|---|
| Frequency-only | 0.625 | 0.375 | 0.080 |
| Semantic distinctiveness-only | 0.100 | 0.900 | 0.257 |
| Minority score (Combined) | 0.200 | 0.800 | 0.150 |

*Table 8.* Neurons are selected by frequency-only (decreasing $\nu_i$), semantic distinctiveness-only (increasing $d_i$), or the combined Minority Score $s(z_i) = d_i(1 - \nu_i)$ (increasing).

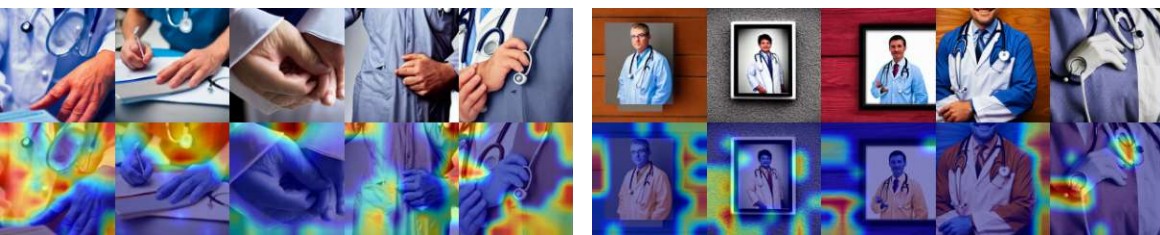

*Figure 11.* (a) Frequency-only minority neuron identification: Neuron with (Left) Frequency 0.01, and (Right) Frequency 0.06. These are the least frequently activated neurons, but appear noisy and uninterpretable.

deem to be uninterpretable and (ii) the mean attribute presence of identified attributes, reported in Table 8.

Table 8 shows that frequency-only selection yields a high uninterpretable rate, revealing a common failure mode of rarity-based ranking in practice: neurons can fire rarely for non-semantic reasons (e.g., dead units or idiosyncratic low-level effects). In a controlled setting where the number of factors is small and the SAE is trained sufficiently well that features align nearly one-to-one with attributes, activation frequency $\nu_i$ is a reliable signal of rarity. In realistic settings, we cannot assume such a perfect feature–attribute alignment, and low activation frequency often reflects sparse, noisy, or semantically inconsistent features rather than genuinely underrepresented attributes. Notably, the mean attribute presence for frequency-only selection is very low, indicating that when a low-frequency neuron is interpretable, it typically corresponds to a genuinely minority attribute, consistent with the controlled experiment, yet some of the neurons remain uninterpretable in practice, as also observed in Figure 11.

Next, we consider the case where neurons are ranked by semantic distinctiveness $d_i$ alone. Table 8 shows that semantic distinctiveness-based selection is highly interpretable. Intuitively, $d_i$ favors neurons that consistently activate on a coherent, non-generic concept whose semantics lie far from the dataset's average representation. When a neuron repeatedly responds to such a specific concept, its top-activating images occupy a stable region in CLIP space, yielding an activation-weighted centroid $\mu_i$ that remains reliably displaced from the dataset centroid $\mu_{D_c}$ and therefore produces a large $d_i$. In contrast, noisy neurons tend to activate on a heterogeneous set of images with no consistent semantics; their CLIP embeddings cancel in different directions, causing $\mu_i$ to drift toward the dataset mean and resulting in a much smaller distinctiveness score. However, semantic distinctiveness alone is insufficient for minority discovery, because deviation from the dataset's average semantics may not always imply underrepresentation. A concept can be coherent and directionally far from the dataset centroid while still being common (i.e., a frequent sub-mode of the distribution). This is reflected in Table 8: although semantic distinctiveness-based selection achieves high interpretability, it yields a higher mean attribute presence, indicating that it can surface semantically coherent yet often prevalent attributes.

These complementary modes motivate the product form $s(z_i) = d_i \cdot (1 - \nu_i)$. As observed in Table 8, the combined criterion preserves most of the interpretability gained by distinctiveness selection while substantially reducing attribute prevalence, and it lowers the fraction of uninterpretable neurons compared to frequency-only selection. While each ablation is optimal on a single axis, the combined score yields the best tradeoff by maximizing the share of interpretable neurons with low attribute presence. Conceptually, $d_i$ acts as a semantic-coherence filter that suppresses rare-but-unstructured activations, whereas $(1 - \nu_i)$ captures underrepresentation and downweights coherent but frequent concepts. Thus, reliably identifying minority-associated neurons requires both signals to avoid selecting either rare noise or common, semantically coherent attributes.

### G.11. Interpretability generalization across prompts

The Minority Score ablation in Appendix G.10 reports interpretability and mean Attribute Presence on a single prompt ("a photo of a doctor," SDXL). To verify that the high interpretability rate of the combined Minority Score is robust across settings, we extend the analysis to four additional WinoBias professions and five COCO prompts, all evaluated on SDXL using the same protocol.

*Table 9.* Interpretability and mean Attribute Presence of RAIGen-discovered minority neurons across additional WinoBias professions, on SDXL. Interpretable rates remain comparable to or higher than the 0.80 reported in Table 8 for "doctor."

| Profession | Interpretable ↑ | Mean Attr. Presence ↓ |
|---|---|---|
| Farmer | 0.80 | 0.251 |
| Sheriff | 1.00 | 0.177 |
| Nurse | 0.80 | 0.229 |
| Attendant | 0.70 | 0.128 |

*Table 10.* Interpretability and mean Attribute Presence of RAIGen-discovered minority neurons across five COCO prompts, on SDXL.

| Prompt | Interpretable ↑ | Mean Attr. Presence ↓ |
|---|---|---|
| A large white building with a big clock tower at one corner. | 1.00 | 0.083 |
| A couple of people carrying surf board walk on a beach. | 0.90 | 0.194 |
| A batter swings the bat as the crowd watches attentively. | 0.80 | 0.085 |
| A young skater is boarding inside of an empty pool. | 0.80 | 0.156 |
| A man talks on his cell phone while he surfs his computer. | 1.00 | 0.210 |

Across all nine additional settings, the interpretable rate ranges from $0.70$ to $1.00$, closely matching the $0.80$ reported in Table 8 for the original "doctor" prompt and exceeding it in seven of nine cases. Mean Attribute Presence is consistently low (between $0.083$ and $0.251$), confirming that the discovered attributes remain genuinely underrepresented across both occupation prompts and open-domain COCO scenes. These results indicate that RAIGen's discovery quality is robust to prompt variations.

### G.12. Sample-complexity study: how many generations are needed?

A practical question for RAIGen is how many generated samples are required before minority-attribute discovery becomes stable. To investigate this, we vary the number of generated samples per prompt $N \in \{100, 1000, 5000, 10000\}$, train an MSAE on the resulting bottleneck representations, and apply the full RAIGen pipeline. We use the prompt "a photo of a doctor" on SDXL and evaluate two complementary properties: (i) the fraction of top discovered neurons that are interpretable, judged by whether the annotation model identifies a consistent semantic attribute across top-activating images (Interpretable Neurons); and (ii) the mean Attribute Presence of identified attributes, where lower presence indicates that the discovered attributes are genuinely rare in default sampling.

*Table 11.* Effect of the number of generated samples per prompt on RAIGen discovery, for the prompt "a photo of a doctor" on SDXL. Discovery becomes stable around $N = 5000$.

| Samples ($N$) | Interpretable Neurons ↑ | Attr. Presence ↓ |
|---|---|---|
| 100 | 0.1 | — |
| 1,000 | 0.5 | 0.41 |
| 5,000 | 0.8 | 0.15 |
| 10,000 | 0.9 | 0.14 |

Table 11 summarizes the results. At $N = 100$, discovery is unreliable since too few neurons are interpretable to compute a stable Attribute Presence estimate. Both metrics improve substantially up to $N = 5000$, after which gains are marginal. Doubling the sample budget from 5,000 to 10,000 improves interpretability from $0.8$ to $0.9$ and Attribute Presence from $0.15$ to $0.14$. We therefore adopt $N = 5000$ throughout the paper as a cost–quality operating point that captures most of the

asymptotic discovery quality at half the compute budget of $N = 10000$. Practitioners with tighter compute budgets can use $N$ as low as 1000 at the cost of reduced interpretability and noisier rarity estimates.

### G.13. Computational cost

We report end-to-end wall-clock time for a single prompt on SDXL, measured on a single NVIDIA A100 (40 GB) GPU. Reported times are for 5,000 generated samples per prompt (the operating point selected in Appendix G.12).

*Table 12.* End-to-end wall-clock time for one prompt on SDXL, single A100 (40 GB).

| Stage | Time |
| --- | --- |
| Image generation + bottleneck extraction (5,000 samples) | 55 min |
| MSAE training + neuron discovery | 16 min |
| LLM annotation of discovered neurons | 44 sec |
| Total per prompt | ∼72 min |

As observed in Table 12, the dominant cost is image generation rather than MSAE training or annotation. This cost is not specific to RAIGen. Any open-set auditing framework that operates on generated samples (e.g., OpenBias (D'Incà et al., 2024)) incurs a comparable cost, since large-scale sampling is unavoidable for identifying systematic failure modes.

## H. Qualitative results

In this section, we present qualitative results illustrating the minority neurons identified by RAIGen across different datasets and prompts. Figures 12 and 13 show minority neurons corresponding to different WinoBias professions from the representations of SDXL. Figures 14 and 15 show minority neurons corresponding to different WinoBias professions from the representations of SD v1.4. Figure 16 displays minority neurons obtained for a set of diverse COCO prompts in SD v1.4. Finally, Figures 17 present the minority neurons identified for the prompt *"A photo of a Doctor"* in SD v2.1. For each neuron, we strongly activating images along with their corresponding activation heatmaps, which highlight the visual features. Since RAIGen surfaces several minority neurons, we present only a representative subset in the figures for clarity and illustration.

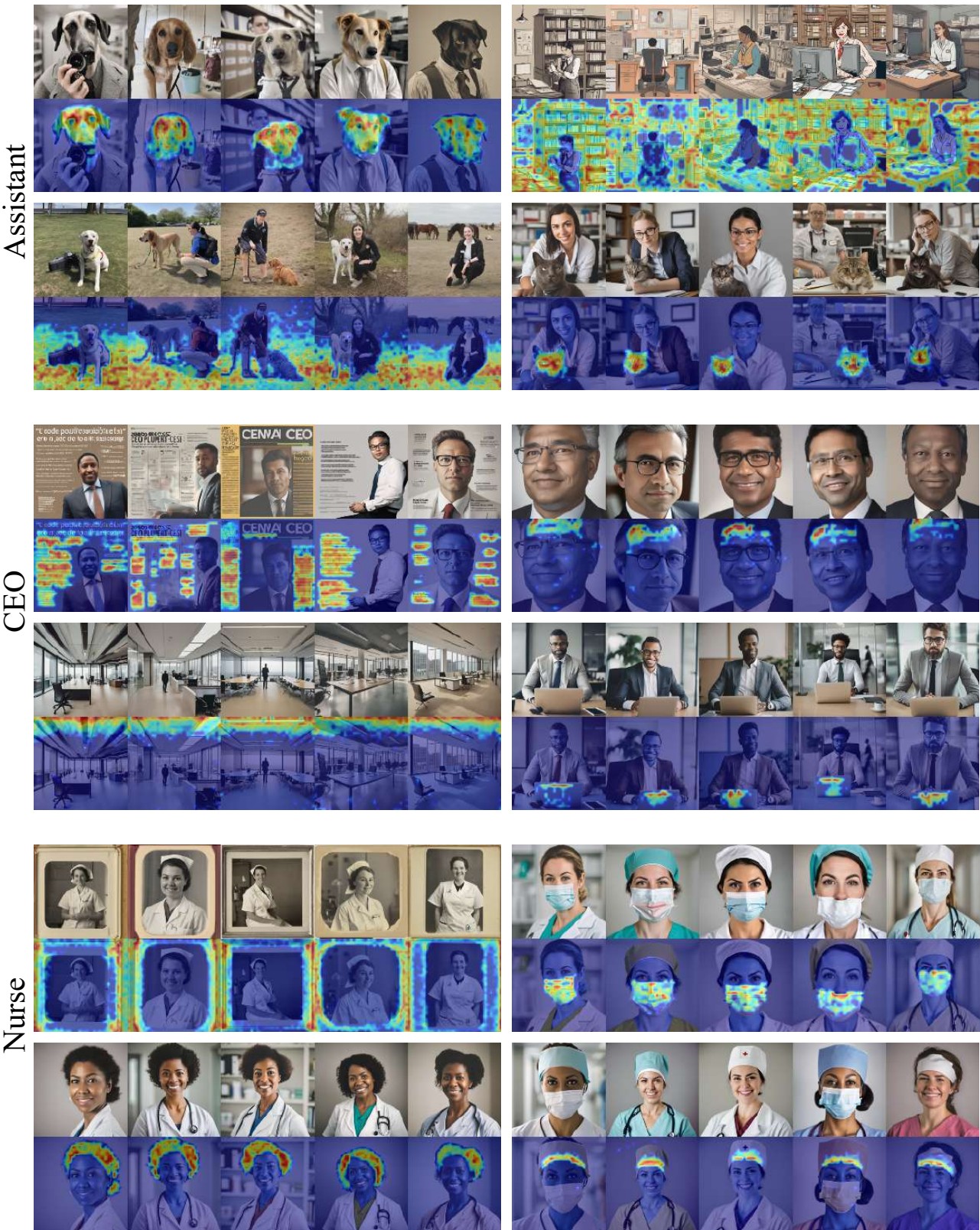

*Figure 12.* Visualization of 4 minority neurons obtained by RAIGen for different WinoBias professions in SDXL. For each neuron, we show the top five activating images and their corresponding activation heatmaps.

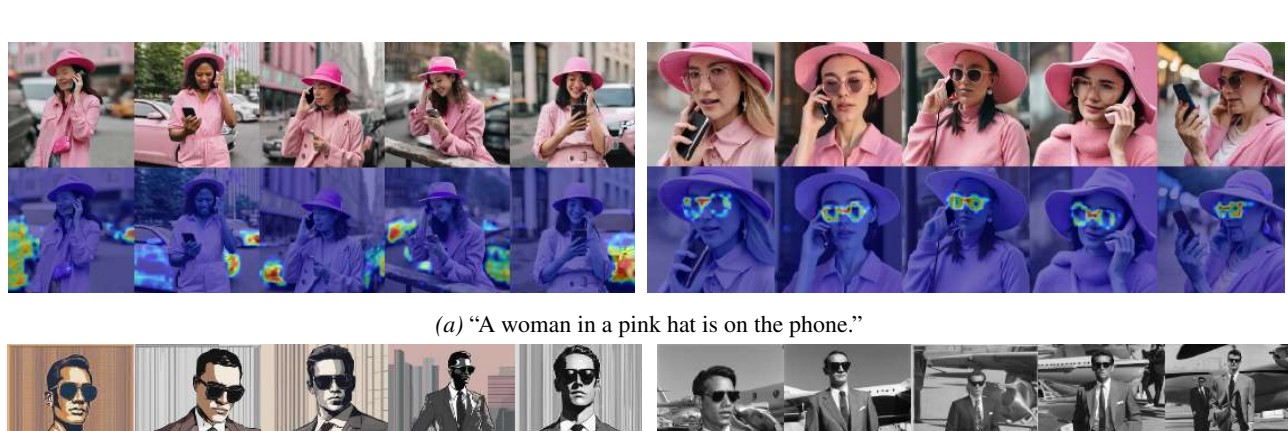

*(a)* "A woman in a pink hat is on the phone."

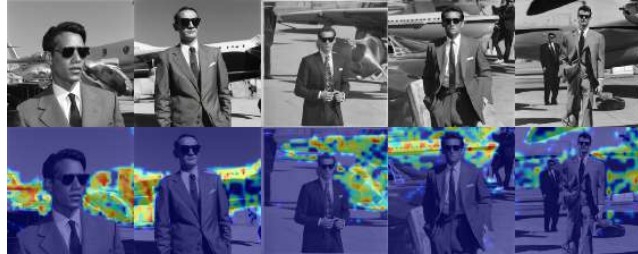

*(b)* "A man in suit and tie is wearing sunglasses."

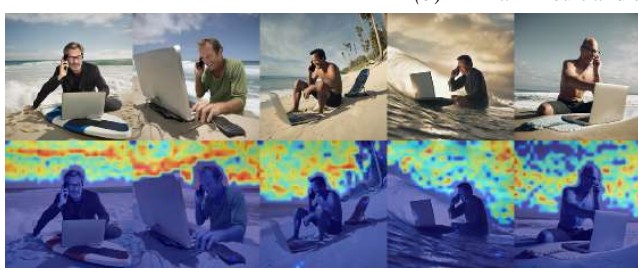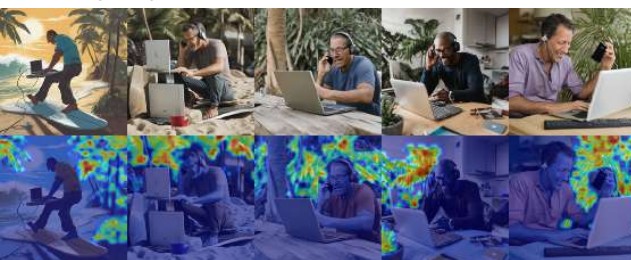

*(c)* "A man talks on his cell phone while he surfs his computer."

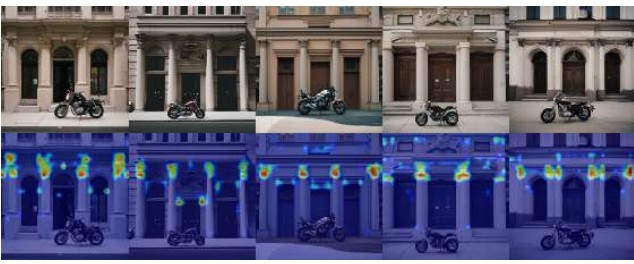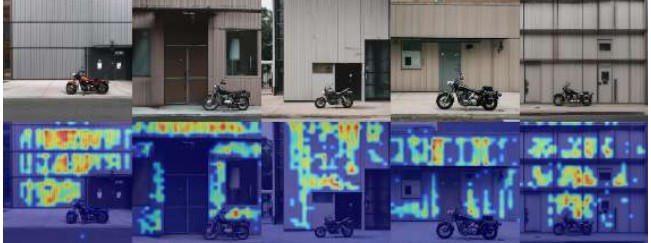

*(d)* "A motorcycle parked outside the doors of a building."

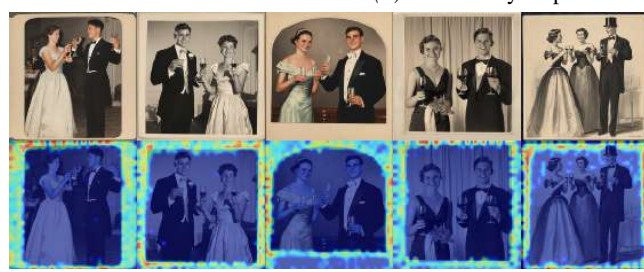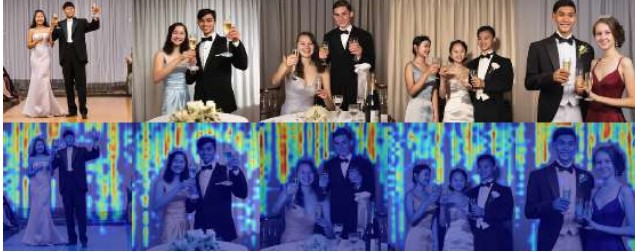

*(e)* "A young couple in formal evening attire toasts the crowd."

*Figure 13.* Visualization of 2 minority neurons obtained by RAIGen for different COCO prompts in SDXL. For each neuron, we show the top five activating images and their corresponding activation heatmaps.

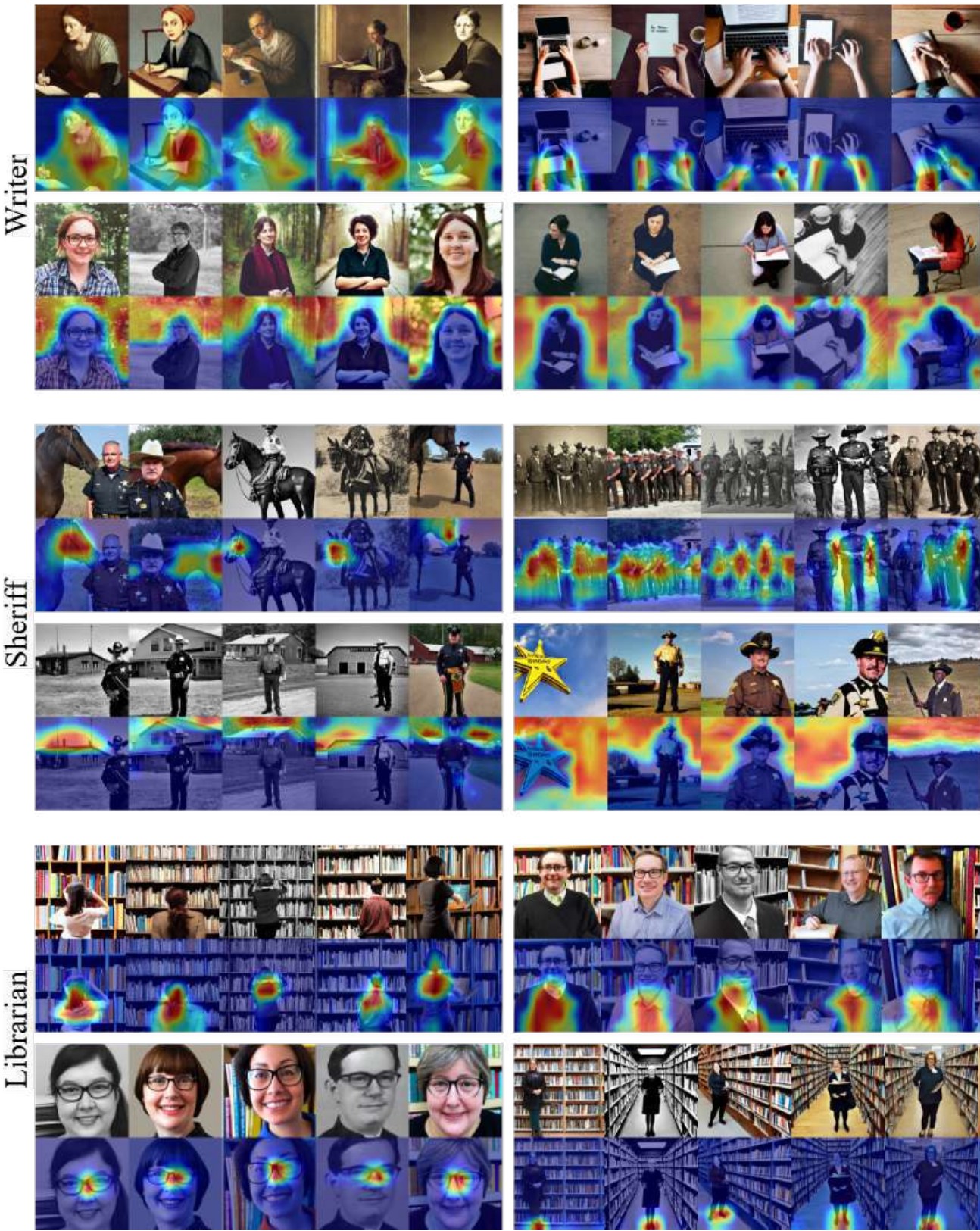

*Figure 14.* Visualization of 4 minority neurons obtained by RAIGen for different WinoBias professions. For each neuron, we show the top five activating images and their corresponding activation heatmaps.

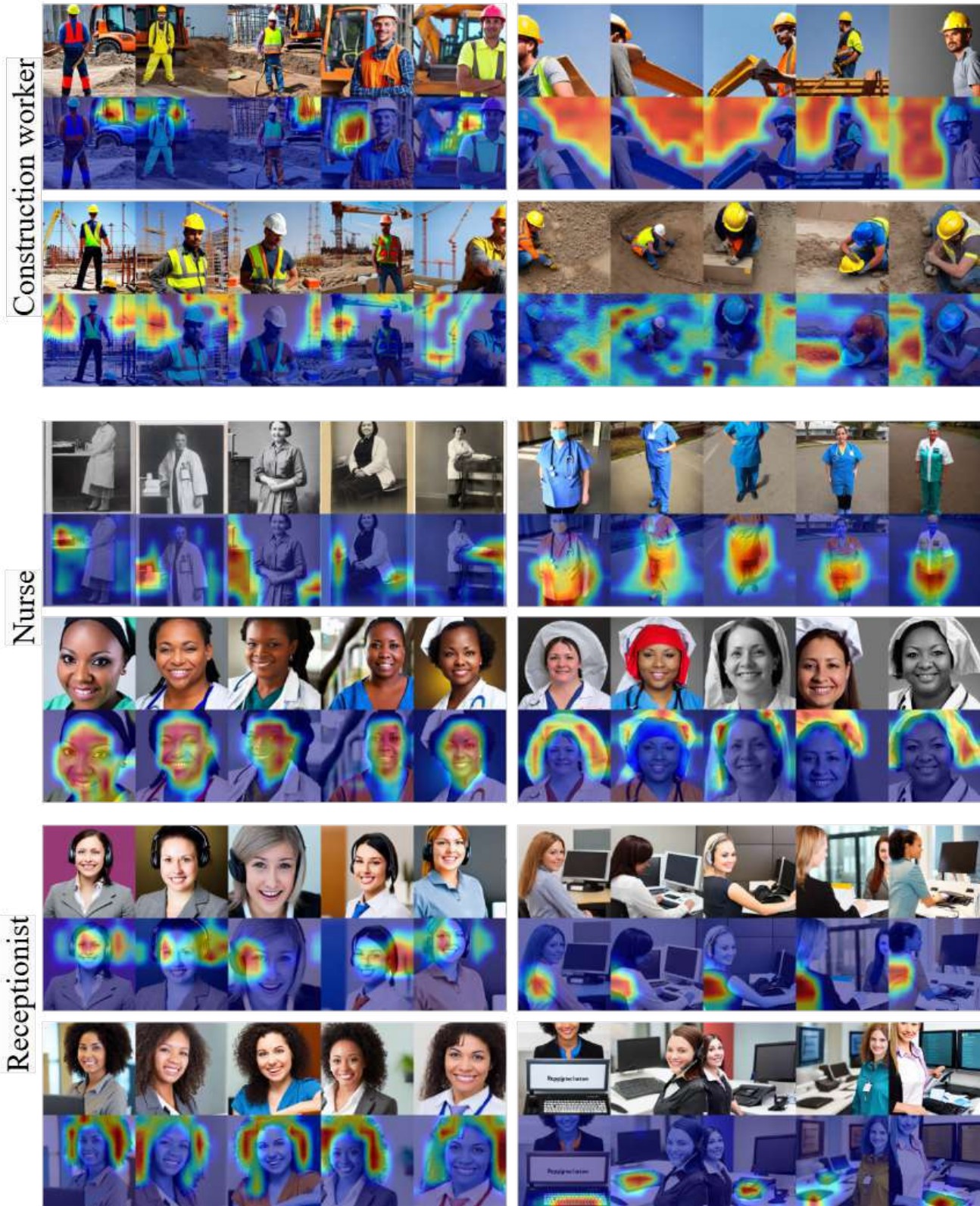

*Figure 15.* Visualization of 4 minority neurons obtained by RAIGen in SD v1.4 for different WinoBias prompts. For each neuron, we show the top five activating images and their corresponding activation heatmaps.

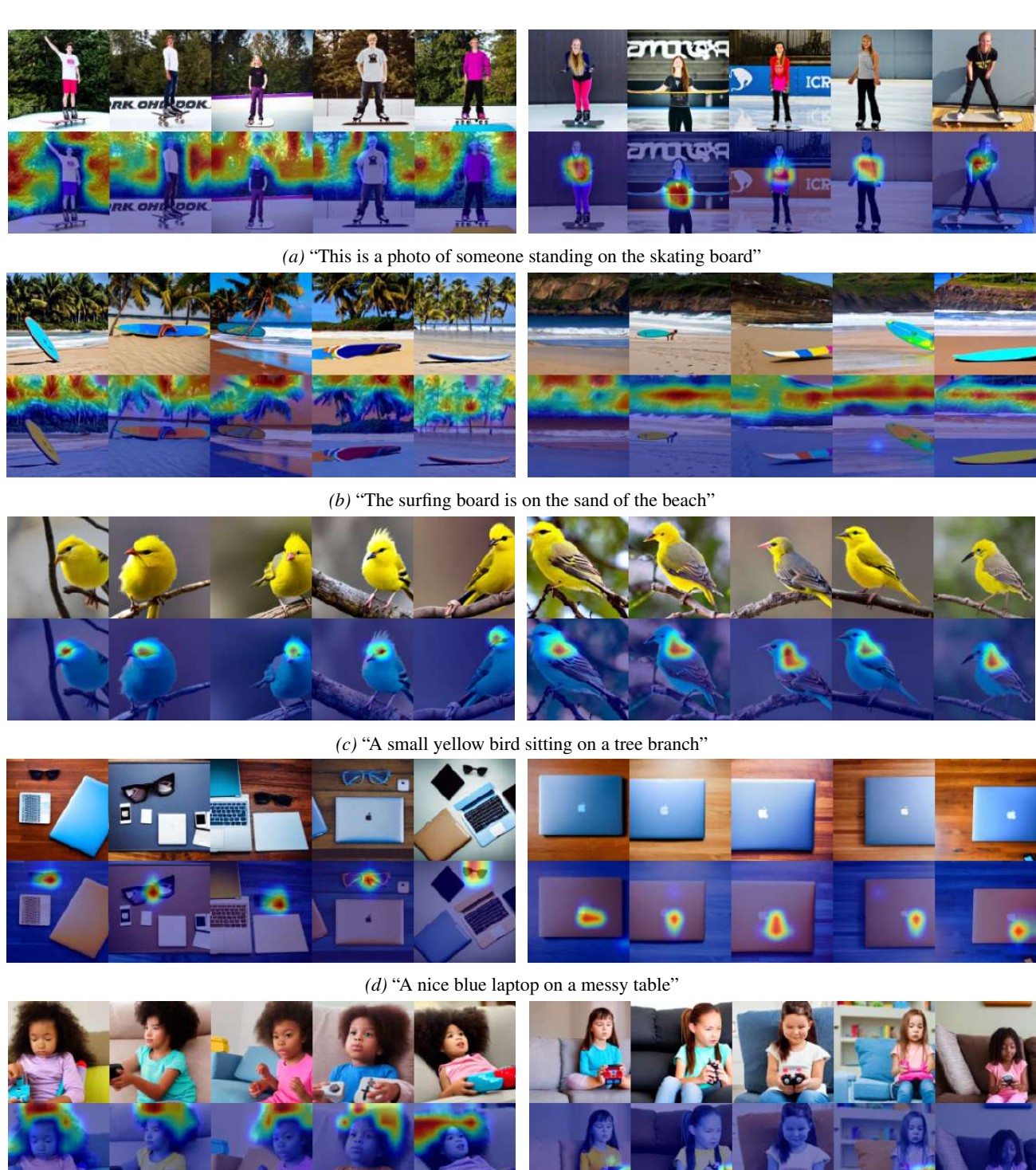

*(a)* "This is a photo of someone standing on the skating board"

*(b)* "The surfing board is on the sand of the beach"

*(c)* "A small yellow bird sitting on a tree branch"

*(d)* "A nice blue laptop on a messy table"

*(e)* "A little girl sitting on the couch playing"

*Figure 16.* Visualization of 2 minority neurons obtained by RAIGen in SD v1.4 for different COCO prompts. For each neuron, we show the top five activating images and their corresponding activation heatmaps.

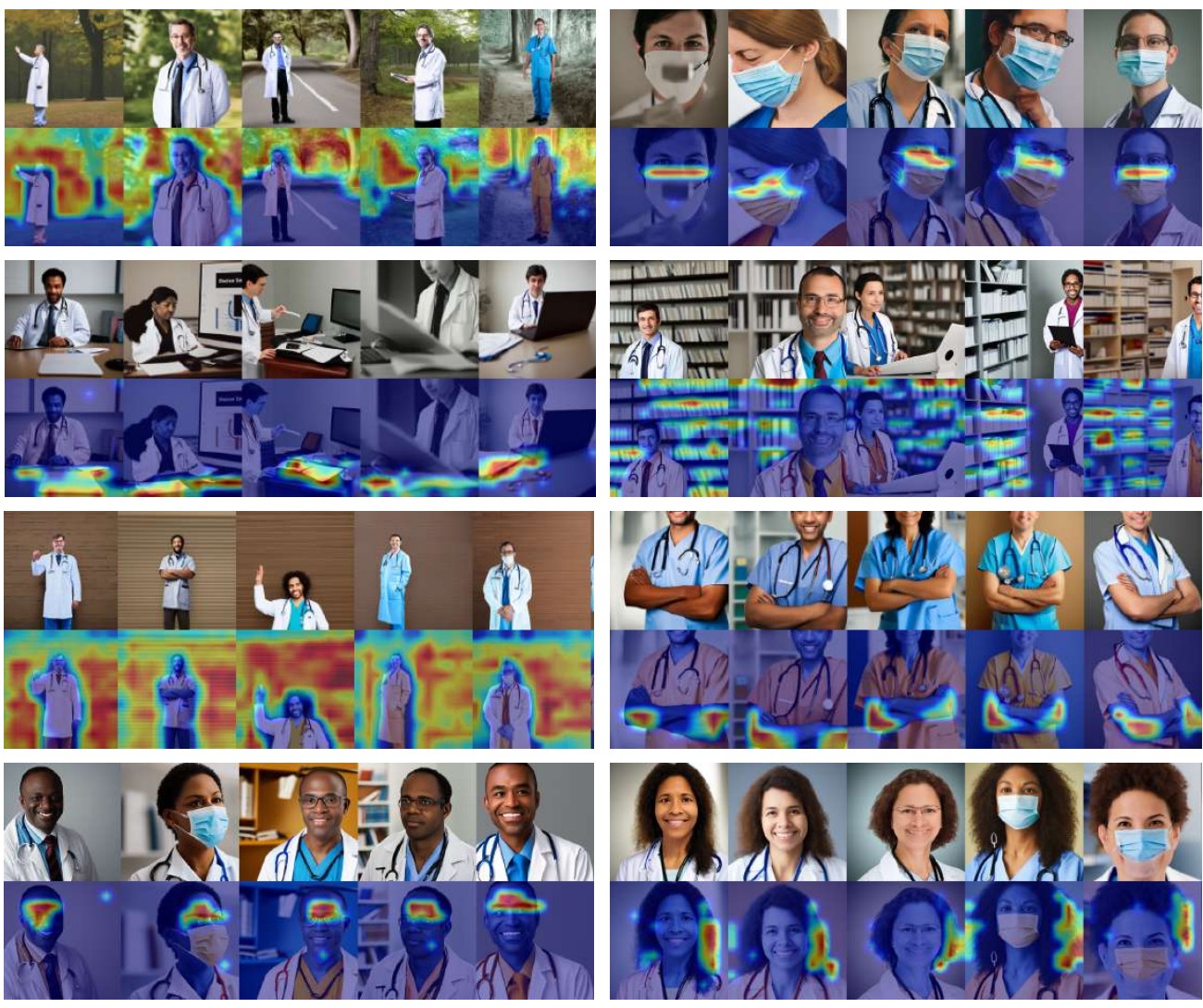

*Figure 17.* Visualization of 8 minority neurons identified by RAIGen for the prompt *"A photo of a Doctor"* in SD v2.1. For each neuron, we show the top five activating images and their corresponding activation heatmaps.

