# OpenReview forum: "RAIGen: Rare Attribute Identification in Text-to-Image Generative Models"
_ICML.cc/2026/Conference — ICML 2026 regular_

### Official Review · Reviewer_LoeD · 2026-03-11

**Soundness:** 2
**Presentation:** 3
**Significance:** 2
**Originality:** 3
**Overall Recommendation:** 4
**Confidence:** 5

**Summary:**

RAIGen is an unsupervised framework that identifies rare attributes encoded within diffusion models but suppressed during generation. To overcome the limitations of prior work focused on predefined fairness categories or dominant majority biases , it introduces a novel metric combining Matryoshka Sparse Autoencoders (MSAE) with activation frequency and semantic distinctiveness. This approach allows for the discovery of hidden minority features in models like SDXL and FLUX and significantly enhances generation diversity by reflecting these attributes in prompts.

**Compliance With Llm Reviewing Policy:**

Affirmed.

**Final Justification:**

Thanks for the detailed rebuttal response from the authors. I maintain my initial positive score.

**Key Questions For Authors:**

- In the Implementation Details, while the GPU specifications and dataset scale for MSAE training are clearly stated, it would be very helpful if the authors could also provide figures for the actual training duration. Additionally, it would be insightful to see the end-to-end inference latency, particularly the additional overhead introduced by LLM calls during the process.

**Limitations:**

yes

**Strengths And Weaknesses:**

**Strength**
- Defining the problem as 'rare attribute discovery' is a fresh perspective.
- Choosing MSAEs to resolve the concept fragmentation issue of standard SAEs is technically sound; specifically, it is designed to capture semantically consistent global attributes rather than local noise by utilizing the coarsest level.
- The Minority Score is also impressive in that it does not simply identify low activation frequency but instead establishes a metric that incorporates semantic uniqueness.
- By applying and validating the method on both UNet and Transformer-based models, the study demonstrates that the proposed approach is universal and not limited to a specific architecture.

**Weakness**
- While the authors claim an unsupervised approach independent of external language models, they actually utilize models such as GPT-5.2 or Llama 4 to label discovered neurons and modify prompts. This contradicts the claim that the entire process was resolved purely through the model's internal data. For the method to be truly unsupervised, shouldn't they have adopted an approach that directly guides the image generation probability distribution using only 'neuron indices' without any assistance from an LLM?
- While the concept of amplifying rare attributes is compelling, it would be very helpful if the authors could provide more reassurance regarding the visual fidelity of the generated images during this process. Specifically, a quantitative evaluation would be highly valuable to confirm that the amplification of these attributes does not lead to any significant degradation in overall image quality.
- Lack of Citation and Competitive Baselines:
1. While this paper compares its performance primarily against OpenBias, a framework designed for identifying majority attributes, this choice of a control group is biased, considering the core objective is minority attribute discovery. Specifically, if minority data generation is sufficiently achievable through sampling via statistical techniques like Minority Guidance [1] and Tweedie’s formula, the MSAEs training costs and LLM-based interpretation required by RAIGen are likely over engineering, where the costs outweigh the practical benefits.
A clear explanation is required to demonstrate the technical superiority of high-cost precision diagnostics (MSAE) over simple sampling guidance.
2. Cones [2] has already demonstrated the existence of 'concept neurons', parameters within the K-V attention layers that determine specific subjects, by utilizing gradient statistics. In contrast to Cones, which achieves efficient customization and high storage efficiency through specific neuron shutting, the paper would be significantly enriched by presenting clear points of differentiation. It would be better if the authors could further clarify the unique strengths of RAIGen’s bottleneck activation analysis, especially in how it offers a distinct perspective on concept identification and control compared to these existing methods.
3. R2F [3] and ToRA [4] have shown high visual fidelity for rare or complex concepts. It would be very insightful to see a comparison showing whether RAIGen's pipeline for amplifying rare knowledge offers a clear qualitative advantage over the more computationally efficient, unsupervised prompt processing seen in methods like ToRA. To further strengthen the paper's impact, we suggest including a quantitative analysis that demonstrates how RAIGen surpasses the generation quality of these existing methods, which already perform well in handling rare attributes.

---

[1] Um, Soobin, Suhyeon Lee, and Jong Chul Ye. "Don't play favorites: minority guidance for diffusion models." arXiv preprint arXiv:2301.12334 (2023).

[2] Liu, Zhiheng, et al. "Cones: Concept neurons in diffusion models for customized generation." arXiv preprint arXiv:2303.05125 (2023).

[3] Park, Dongmin, et al. "Rare-to-frequent: Unlocking compositional generation power of diffusion models on rare concepts with llm guidance." arXiv preprint arXiv:2410.22376 (2024).

[4] Kang, Seil, et al. "Rare Text Semantics Were Always There in Your Diffusion Transformer." arXiv preprint arXiv:2510.03886 (2025).

---

> ### Author Rebuttal · Authors · 2026-03-30
>
> We thank the reviewer for their thoughtful feedback. We would like to clarify a fundamental distinction: **RAIGen is a diagnostic and auditing framework**, not a method that improves rare generation. Its primary purpose is to discover attributes that are encoded within diffusion models but suppressed. The amplification experiment (Sec. 5.4) serves only as a downstream validation of actionability and practical utility. This distinction addresses several of the reviewer’s concerns, detailed below.
>
> **W1.** The core discovery pipeline (MSAE, Minority Score) operates without labels, predefined categories, or LLMs. Minority attributes are identified purely from the diffusion model's internal representations. LLMs serve only for post-hoc text annotation of already-discovered neurons, analogous to labeling clusters after unsupervised clustering. Neurons are independently interpretable via top-activating images and heatmaps (Fig. 3, 4). We acknowledge, as Reviewer wn22 (W1) also notes, that CLIP used for distinctiveness constitutes a pretrained semantic prior, though not supervision over which attributes are rare. We will clarify this by describing RAIGen as label-free, requiring no predefined categories, while using LLMs only for annotation. **Neuron-level steering:** Direct neuron-level interventions are a natural alternative to prompt-based amplification and are compatible with RAIGen, as prior SAE-based steering methods (Cywiński & Deja, 2025) demonstrate. We use prompt revision as a lightweight demonstration of actionability, since our focus is discovery rather than generation. However, for settings like cross-model auditing (App. F.3), human-readable LLM annotations provide a convenient interface. The downstream method is application-dependent, while RAIGen’s core discovery remains unchanged.
>
> **W2.** CLIP prompt-image alignment is reported in Tables 3–4. Following the suggestion, we compute CLIP-IQA averaged over 10 WinoBias professions on SD v1.4:
> | Metric | Base Prompt | RAIGen-Revised |
> | -------- | -------- | -------- |
> | CLIP-IQA | 0.6932 | 0.6891 |
>
> Both show negligible degradation. However, improving rare generation quality is not the focus of our method. Higher-fidelity rare generation can be achieved by pairing RAIGen with dedicated methods, which we leave for future work.
>
> **W3.1. Minority Guidance:** Minority Guidance and RAIGen address distinct tasks. They steer generation toward low-density regions but do not identify which attributes are rare or merit amplification. We validate this experimentally: generating SDXL samples for “A photo of an assistant,” RAIGen identifies animals (e.g., cats, dogs) as rare attributes. We also generate rare samples by applying Minority Guidance on SDXL on the same prompt. We compute Attribute Presence for “animals” for both SDXL and Minority Guidance:
>
> | Method | Attribute Presence metric for '"animals"
> | -------- | -------- |
> | Base - SDXL     | 0.150    |
> | Minority Guidance | 0.382 |
>
> Minority Guidance amplifies “animals” due to statistical rarity, but this attribute is contextually irrelevant for a profession task. This addresses the overengineering concern: without identifying what is rare and meaningful, sampling-based guidance cannot distinguish actionable minorities from noise, making RAIGen a necessary prerequisite rather than redundant overhead. The two operate at different stages: RAIGen identifies rare attributes, while Minority Guidance amplifies any satisfying its heuristic, regardless of relevance. OpenBias remains the appropriate comparison as the only open-set attribute discovery method.
>
> **W3.2. Cones:** Cones identify concept neurons for user-specified subjects via gradient statistics in K–V attention layers. RAIGen differs fundamentally by discovering unknown minority attributes without user specification or reference images. Cones answers “how to generate a concept,” whereas RAIGen answers “what concepts are learned but suppressed.” RAIGen operates on UNet bottleneck representations, shown by Kwon et al. (2023) to be inherently semantic, capturing holistic scene attributes. MSAE decomposition yields a comprehensive set of factors at controllable granularity, enabling systematic rarity quantification, which Cones is not designed for.
>
> **W3.3. R2F and TORA:** Both are generation methods for user-specified rare concepts and assume prior knowledge of which concepts to generate. RAIGen addresses the upstream problem of discovering suppressed attributes, making direct generation-quality comparison not directly applicable. The approaches are complementary: RAIGen provides the missing prerequisite, identifying what to generate, which these methods do not. Its outputs can serve as inputs to R2F or ToRA for higher-fidelity amplification. As noted in W2, we leave this to future work.
>
> **Q1.** Please refer to the response for Reviewer WB4Q (m3).

---

> > ### Author Rebuttal · Reviewer_LoeD · 2026-04-04
> >
> > I appreciate the authors' thorough rebuttal. The clarification on the label-free framing (W1), the CLIP-IQA results (W2), and the Minority Guidance experiment (W3.1) are convincing and address my main concerns.

---

> > > ### Author Response · Authors · 2026-04-08
> > >
> > > We sincerely thank the reviewer for the positive acknowledgement and for confirming that the concerns have been fully addressed. We hope the reviewer will consider updating the score accordingly. We appreciate the constructive feedback, which has helped strengthen the paper.

---

### Official Review · Reviewer_uoTc · 2026-03-13

**Soundness:** 3
**Presentation:** 3
**Significance:** 2
**Originality:** 3
**Overall Recommendation:** 4
**Confidence:** 4

**Summary:**

This paper proposes RAIGen, a method for unsupervised rare attribute discovery in diffusion models. RAIGen first trains Matryoshka Sparse Autoencoders (MSAEs) on the h-space of diffusion models, and then computes a minority score in the learned sparse representation space. This score is designed to capture both rarity, based on activation frequency, and semantic distinctiveness, based on CLIP representations of generated samples. The paper further demonstrates the interpretability of the discovered attributes through automated annotation. Experiments on Stable Diffusion, SDXL, and FLUX.1 suggest that RAIGen is scalable across different diffusion architectures and can identify rare attributes in multiple models.

**Compliance With Llm Reviewing Policy:**

Affirmed.

**Final Justification:**

I appreciate the authors’ detailed rebuttal and discussion. Based on the clarifications, I will raise my score to weak accept, as I am now less concerned about the issues regarding practical and social significance.

I believe this direction could provide useful inspiration to the community. That said, additional evidence would be helpful to more clearly demonstrate when the discovered attributes translate into practically useful insights and socially meaningful bias analysis, rather than simply reflecting general long-tailed generation behavior.

**Key Questions For Authors:**

- In Definition 1, should the construction of the attribute set $A$ be subject to additional constraints? For example, should attributes in $A$ be independent of the conditioning variable $c$, or should $A$ follow a more structured definition?
- The presentation of Table 1 is somewhat ambiguous. Although Attribute Presence is defined as the fraction of images containing a given attribute, what exactly is being compared between the majority fraction and minority fraction? Also, what does the $\downarrow$ notation specifically refer to here?
- Since the paper introduces rare attribute identification in diffusion models as a new task, I do not think the current experiments are sufficient to fully establish the task’s necessity and practical value. For instance, can RAIGen be used to systematically evaluate the extent of minority bias in a given diffusion model, or can it be further leveraged to mitigate such underrepresentation bias?

- More importantly, some of the discovered rare attributes currently seem of limited significance. Examples such as “writer with hat” or doctor in a specific background appear closer to prompt underspecification or sampling variation than to genuinely overlooked minority attributes. This raises the question of whether RAIGen is discovering socially or semantically meaningful rare attributes, or merely identifying low-frequency but weakly informative visual patterns. Is RAIGen discovering socially or semantically meaningful rare attributes, or mainly low-frequency but weakly informative visual patterns?

**Limitations:**

Yes

**Strengths And Weaknesses:**

**Strengths:**
- The paper is well motivated and studies an important problem: the unsupervised discovery of minority attributes in generative models in a way that is interpretable to humans.
- The related work section on debiasing is comprehensive, and the paper is generally well organized. The problem formulation and key notions are presented clearly.


**Weakness:**
- The experimental evaluation is currently not sufficient to fully support the paper’s claims. Please see the questions for more details.
- The overall pipeline appears relatively heavy, as it depends on training MSAEs separately for specific diffusion models. This may limit its practicality and scalability in broader settings.
- The methodological novelty also seems somewhat limited, since SAE-based approaches for debiasing or responsible generation in diffusion models have already been explored in prior work, such as:
[1] Dissecting and Mitigating Diffusion Bias via Mechanistic Interpretability (CVPR 2025)
[2] SAeUron: Interpretable Concept Unlearning in Diffusion Models with Sparse Autoencoders (ICML 2025)
- The experimental section is not very well organized, and several figures are difficult to read. For example, the VLM-generated annotations can easily be mistaken for generation prompts (Fig. 3, Fig.4). In addition, Section 5.1 reads more like a pilot study or preliminary analysis and might fit better earlier in the paper, possibly in Section 4. More broadly, leaving many important details to the appendix further hurts readability.
- There are also some issues in presentation. For example, regarding the automated annotation, the paper appears to use LVLMs/MLLMs rather than LLMs as annotators (Line 274).

---

> ### Author Rebuttal · Authors · 2026-03-30
>
> We thank the reviewer for the valuable feedback. We address each of the concerns below:
>
> **W2.** Please refer to the response for Reviewer WB4Q (m3).
>
> **W3.** Both cited works operate on predefined bias categories. DiffLens identifies and mitigates known social biases such as gender, race and age. SAeUron uses SAEs for concept unlearning of known target concepts. RAIGen addresses a fundamentally different task: **open-set discovery of unknown minority attributes without predefined categories.** The novelty lies not in using SAEs on diffusion models, but in (i) the formulation of rare attribute discovery as a new task, (ii) the Minority Score combining activation frequency with semantic distinctiveness.
>
> **W4,5.** We thank the reviewer for the feedback. We will address the presentation concerns in the final version, including clarifying the annotations, improving figure readability, and reorganizing sections.
>
> **Q1.** In our formulation, $A$ is conditioned on $c$, as the relevant attribute space varies across prompts. What constitutes a minority attribute for "Doctor" differs from "Sheriff." Imposing independence constraints between $A$ and $c$ would preclude discovering prompt-specific minority attributes, which is RAIGen's core objective and the setting we operate in throughout the paper. However, RAIGen is also compatible with a prompt-independent setting. If MSAE is trained on representations pooled from a diverse set of prompts, the discovered minority attributes would reflect globally underrepresented attributes across the entire generative distribution. The framework naturally accommodates both settings depending on how the MSAE is trained. The Minority Score computation remains identical. We will clarify this flexibility in the final version.
>
> **Q2.** We appreciate this feedback. Attribute Presence measures the fraction of generated images containing a given attribute. OpenBias targets majority attributes (high presence), while RAIGen targets minority attributes (low presence). The ↓ notation is potentially confusing, as it conflates these opposite directions in a single column. We will revise the table to separate majority vs. minority attribute presence and clarify the notation.
>
> **W1/Q3.** We respectfully clarify that our paper already demonstrates both systematic evaluation and mitigation across multiple experiments.
> 1. Systematic evaluation: Appendix F.3 uses RAIGen as a cross-model auditing framework to understand how minority attributes evolve across different diffusion architectures (SD v1.4, v2.1, SDXL). RAIGen provides practitioners with comparisons and actionable insights into representational gaps across model versions.
> 2. Mitigation: Section 5.4 and Appendix F.5 demonstrate that RAIGen's discovered attributes can be directly leveraged to mitigate underrepresentation bias. RAIGen-guided prompt revision increases the coverage of previously suppressed minority modes in generated outputs (Tables 3, 4), providing quantitative evidence of improvement.
> 3. Selective control for rare generation: As demonstrated in our response to Reviewer LoeD (W3.1), RAIGen provides a critical control layer by surfacing interpretable minority attributes, enabling practitioners to make informed decisions about which attributes warrant amplification, a capability that existing rare-generation approaches fundamentally lack. We expand on RAIGen's broader practical applications, including its role as a missing prerequisite for targeted rare-generation pipelines like R2F, in our response to Reviewer Wn22 (W3).
>
> We believe these experiments collectively establish the task's necessity and RAIGen's practical value for auditing, mitigation, and informed decision-making across models and datasets.
>
> **Q4.** When a prompt like "a photo of a doctor" is given as input to the model, the model chooses how to fill in unspecified attributes. These decisions are not random and the model systematically favors certain modes (e.g., male, clinical office) over others (e.g., female, outdoor setting) across thousands of generations. This systematic suppression is precisely what constitutes bias, and RAIGen surfaces these suppressed modes. That these are genuine patterns rather than sampling variation is evidenced by their appearance as coherent MSAE neuron clusters with spatially localized heatmaps (Figures 3, 4). Whether a discovered attribute is "meaningful" depends on the application context. For fairness auditing, demographic minorities (female doctor, Black skin tone, which RAIGen discovers, Fig. 3, Appendix F.3) are most relevant. For creative platforms seeking diverse imagery, stylistic and contextual attributes like "writer with hat" are equally valuable. RAIGen's strength is that it does not predetermine which forms of underrepresentation matter. It surfaces the complete landscape of suppressed attributes, empowering practitioners to decide which warrant intervention. This is a fundamental advantage over closed-set approaches.

---

> > ### Author Rebuttal · Reviewer_uoTc · 2026-04-03
> >
> > **Q1 & Q3.** The practical value is still questionable. Many of the identified attributes (e.g., clinic/style/gesture for the prompt “doctor”) may simply reflect natural semantic priors rather than meaningful bias. In practice, when users require specific scenarios (e.g., outdoor settings or particular styles), they can directly specify these attributes in the prompt. This raises the question of whether discovering such “minority attributes” provides additional practical benefit beyond standard prompt specification.
> >
> > For **Q4**, the results on gender/race mainly serve to validate the method on known bias categories. The key contribution should instead lie in discovering previously unknown attributes and providing deeper analysis of them. At present, I am still not fully convinced that the paper sufficiently distinguishes socially or semantically meaningful minority attributes from general long-tailed generation patterns.
> >
> > Overall, I find the direction interesting, but I remain unconvinced that the current experimental evidence fully supports the claimed task formulation and its broader practical significance. Therefore, I keep my score, while remaining open to acceptance.

---

> > > ### Author Response · Authors · 2026-04-05
> > >
> > > We thank the reviewer for their continued engagement and openness to acceptance. We address Q1/Q3 and Q4 together, as they share a common thread.
> > >
> > > **On whether suppressed attributes merely reflect semantic priors (Q1/Q3)**. We acknowledge that some RAIGen discoveries align with what one might predict (e.g., outdoor doctors). However, we highlight that many discoveries are genuinely unpredictable:
> > >
> > > * For "A photo of an assistant" (Fig. 12, SDXL), RAIGen discovers animals (cats, dogs) alongside human assistants which is a learned association no semantic prior would anticipate.
> > > * For "A young woman poses comically with a piece of pizza", RAIGen discovers afro-curly textured hair as suppressed. There is no semantic link between pizza and hair texture.
> > > * For "A small yellow bird on a tree branch" (Fig. 16c), RAIGen discovers suppressed species diversity. This is a meaningful bias.
> > >
> > > We believe that a practitioner would have no reason to suspect these specific suppressions without RAIGen.
> > >
> > > **On practical value beyond prompt specification (Q1/Q3)**. The reviewer correctly notes that once an attribute is known, specifying it in the prompt is straightforward. But this is precisely RAIGen's contribution: identifying which attributes require specification. Every unspecified dimension in a prompt is a dimension where bias operates silently which is the setting in which all open-set bias auditing operates, including RAIGen. Methods like OpenBias (D'Incà et al., 2024) operate in exactly the same unspecified space, because that is where the model imposes its learned defaults. The difference is that prior work identifies which defaults dominate and RAIGen identifies which alternatives are suppressed. Prompting "a female doctor" resolves gender but leaves skin tone, hair texture, clothing, and setting unspecified, and each specification opens new unspecified dimensions. The space of possible suppressions is therefore too large or infinite for manual inspection. Without RAIGen, practitioners have no systematic way to know which of these dimensions are biased and which are not. This matters at deployment scale. Existing rare-generation methods such as R2F (Park et al., 2024) and ToRA (Kang et al., 2025) require knowing what is rare as input. RAIGen provides the upstream discovery step that these methods need but lack.
> > >
> > > **On distinguishing meaningful discoveries from long-tail noise (Q4)**. We appreciate this concern. We find it helpful to note a key asymmetry. In discriminative long-tail recognition, identifying rare classes is straightforward where we count labels. In generation, there are no predefined categories over which to measure imbalance. A "class" in the generative tail could be either a viewpoint, a species, a hair texture which is itself unknown. Every example above : animals as assistants, afro-curly hair on a pizza prompt, species collapse for yellow birds was unknown before RAIGen discovered it and they are meaningful discoveries. Whether a given discovery warrants intervention is inherently application-dependent: suppressed species diversity matters for an educational platform but not for a recruitment tool. Suppressed hair textures matter for a stock photography service but not for an autonomous driving pipeline. RAIGen's role is to surface the complete inventory of suppressed modes. The practitioner provides the domain-specific judgment on which entries matter. We will add a discussion making this framing explicit in the revised paper.
> > >
> > > We are grateful for the reviewer's constructive engagement throughout this process.

---

### Official Review · Reviewer_WB4Q · 2026-03-13

**Soundness:** 3
**Presentation:** 3
**Significance:** 3
**Originality:** 3
**Overall Recommendation:** 4
**Confidence:** 3

**Summary:**

This paper formulates a new problem in the bias landscape of text-to-image diffusion models: the unsupervised discovery of minority attributes that are encoded in the model's internal representations but systematically underexpressed during generation. While closed-set methods mitigate bias along predefined fairness axes (e.g., gender, race) and open-set methods like OpenBias surface majority attributes that dominate outputs, neither explicitly targets the identification of rare, suppressed attributes. RAIGen fills this gap by training Matryoshka Sparse Autoencoders (MSAEs) on diffusion UNet bottleneck representations and ranking coarse-level neurons via a Minority Score that multiplies activation rarity with CLIP-based semantic distinctiveness. The framework is validated on a synthetic toy task (Spearman $\rho \approx 0.991$ over 20 seeds), applied to WinoBias (10 professions) and COCO (50 prompts) with SD v1.4 and SDXL, extended to SD v2.1 and FLUX.1-schnell in the appendix, and complemented by a 25-participant user study, a prompt-revision amplification experiment, and three ablation studies (score components, MSAE vs.\ vanilla SAE, coarse-level size).

**Compliance With Llm Reviewing Policy:**

Affirmed.

**Final Justification:**

Novel task formulation with a technically sound framework across four architectures. The rebuttal addressed my main concerns: M1 extended to 9 settings (interpretability 0.70–1.00), M2 extended to 6 professions with consistent RAIGen advantage. Annotation pipeline and cost clarified. Inter-rater reliability unreported and m1/m4 remain revision promises, but these are minor gaps. Rebuttal reinforced my prior assessment. I maintain 4 (Weak Accept).

**Key Questions For Authors:**

1. **Interpretability analysis generalization (M1).** Table 6's interpretable/uninterpretable breakdown is limited to a single prompt and model. Can you extend this analysis to at least 3–5 additional WinoBias professions and a few COCO prompts? If the 80% interpretability rate holds across settings, this would significantly increase my confidence in the method's robustness and I would consider raising my score.

2. **Direct baseline comparison (M2).** Have you compared RAIGen against simpler image-level strategies — for instance, $k$-means clustering on CLIP embeddings of generated images, with small clusters treated as minority candidates? Even a focused comparison on a few prompts showing that RAIGen discovers attributes missed by clustering (or provides more interpretable/coherent results) would strengthen the evaluation.

3. **Annotation pipeline clarification (m2).** The main text credits GPT-5.2 for neuron annotation while the appendix presents a Llama 4-Scout prompt. Could you clarify which model produced the final reported annotations and confirm that annotation and evaluation used separate models?

4. **Computational cost (m3).** What is the approximate wall-clock time and GPU requirement for the full RAIGen pipeline on a single prompt?

**Limitations:**

Yes.

**Strengths And Weaknesses:**

## Strengths

**S1. [Originality / Significance] The problem formulation is genuinely novel and empirically motivated.**
The distinction between "majority bias detection" and "minority attribute discovery" is not merely semantic. The authors substantiate its necessity in Appendix F.1, where suppressing the dominant attribute (White) in Doctor/Manager prompts frees probability mass that shifts disproportionately toward one minority group (Black) while leaving others (East Asian) largely unchanged. This directly demonstrates that majority suppression does not uniformly amplify minorities, justifying the need for dedicated minority discovery. The formal treatment (Definitions 1–2) is clean, and the $P_G(a_j \mid c) > 0$ condition honestly bounds the method's scope to attributes already encoded by the model — a constraint the authors carry through consistently to their Limitations section.

**S2. [Soundness] The toy experiment provides principled grounding.**
Section 5.1 replicates Bussmann et al.'s hierarchical data generation and shows that least-active MSAE latents preferentially align with rare ground-truth features (Figure 2), with a near-perfect rank correlation ($\rho \approx 0.991$). Crucially, the authors do not over-generalize: they frame this as validation under near-ideal alignment and immediately point to Appendix F.8 to show that frequency alone is noisy in practice (62.5% uninterpretable at frequency-only selection), motivating the combined score. This validate-then-qualify structure reflects careful experimental design.

**S3. [Soundness] The ablation studies are multi-layered and informative.**
Three independent ablations each justify a different design decision:
(i) *Score components* (Table 6): frequency-only surfaces rare but largely uninterpretable neurons (62.5% uninterpretable, presence 0.080); distinctiveness-only is interpretable but captures common attributes (presence 0.257); the combined score achieves the best trade-off (20% uninterpretable, presence 0.150).
(ii) *MSAE vs.\ vanilla SAE* (Appendix F.6): standard SAEs fragment concepts — e.g., splitting left eye and right eye into separate neurons — producing misleading minority candidates. MSAE's coarse level mitigates this.
(iii) *Coarse level size* (Table 5): $k_1 = 2048$ recovers key demographic attributes without the over-fragmentation observed at $k_1 = 10240$.

**S4. [Significance] The cross-model analysis yields independently valuable insights.**
Figure 8 tracks minority attribute presence across SD v1.4, v2.1, and SDXL for four attribute categories (demographic, stylistic, contextual, gestural). The finding that demographic balance improves with scale while stylistic diversity (sepia tone, vintage clothing) and contextual diversity (bookshelves, hospital corridors) decrease — a phenomenon the authors term "contextual homogenization" — is a concrete, actionable insight for model developers and a reusable auditing framework.

**S5. [Soundness] The user study provides independent human validation.**
Twenty-five participants confirmed that RAIGen-discovered attributes appear in fewer than 3 out of 10 images on average across all five professions (Table 2), with the strongest scarcity for CEO (0.70/10). This constitutes perceptual evidence that the discovered attributes are not just statistically rare in representation space, but also visibly underexpressed in generated images.

**S6. [Presentation] Reproducibility is well-supported.**
Appendix D provides MSAE hyperparameters, image counts, CLIP model specifications, pruning thresholds, and the full annotation prompt, covering nearly all information needed for replication.

## Weaknesses

### Major

**M1. [Evaluation] The interpretability analysis is limited to a single prompt and model.**
Table 6's breakdown of interpretable vs.\ uninterpretable neurons (80% interpretable for the combined score) is conducted only on "A photo of a doctor" with SDXL. This is the main quantitative evidence that the method discovers *meaningful* rare attributes rather than noise, yet it covers a single setting. Extending this analysis across multiple professions and COCO prompts would substantially increase confidence that the 80% rate is robust. Similarly, the user study covers five professions but reports no inter-rater reliability metric (e.g., Krippendorff's $\alpha$), making it difficult to assess judgment consistency across participants.

### Moderate

**M2. [Evaluation] No direct comparison with simpler rare-attribute discovery strategies.**
The paper compares against OpenBias, which targets majority attributes by design, making the Attribute Presence gap in Table 1 partly expected. A comparison with image-level strategies (e.g., CLIP-embedding clustering or outlier detection) would more directly test whether neuron-level decomposition is necessary for this task. The paper does provide indirect evidence of unique value — spatial heatmaps (Figures 3–4), cross-model neuron tracking (Figure 8), and resistance to concept fragmentation (Appendix F.6) — which simple clustering cannot easily replicate. Given that this is the first work to formulate this problem and no existing baselines target the same task, this is a limitation on demonstrated scope rather than a fundamental gap, but a direct comparison would meaningfully strengthen the contribution.

### Minor

**m1. [Evaluation] Amplification results do not fully isolate RAIGen-specific contributions.**
The deviation ratio improvement ($0.50 \to 0.22$ on WinoBias) is encouraging, but the paper does not separate RAIGen-specific contributions from the general effect of injecting rare descriptors into prompts. A comparison with descriptors from a non-RAIGen source (e.g., human-written rare descriptions) would help isolate the framework's added value.

**m2. [Presentation] The annotation model description is ambiguous.**
Section 5.2 and Appendix D.1 state that minority neurons are annotated with GPT-5.2, but the appendix also presents a prompt formatted for Llama 4-Scout. It appears that GPT-5.2 produced the final annotations while Llama 4-Scout + Qwen3-VL served as the independent evaluation ensemble, which would preserve evaluation independence. However, the current text does not make this fully clear, and a brief clarification would remove any ambiguity.

**m3. [Reproducibility] Computational cost is not reported.**
Each prompt requires generating 10,000 images and training a prompt-specific MSAE. The wall-clock time and GPU requirements for the 60-prompt experimental suite are not provided, making it difficult to assess practical feasibility as an auditing tool.

**m4. [Methodology] No comparison of alternative Minority Score functional forms.**
The element-wise product in Eq.\ 4 is intuitive and the ablation in Table 6 demonstrates that both components are necessary, but no alternative combination strategies (e.g., $\alpha d_i + \beta(1 - \nu_i)$) are tested. A brief sensitivity analysis would strengthen the robustness of this contribution.

---

> ### Author Rebuttal · Authors · 2026-03-30
>
> We thank the reviewer for the valuable feedback and thoughtful comments. We address each of the concerns below:
>
> **M1/Q1.** As per the reviewer's suggestion, we extend the interpretability analysis on additional WinoBias professions and COCO prompts on SDXL, and the results are summarized below.
>
> | Profession | Interpretable ↑ | Mean Attr. Presence ↓ |
> |------------|-----------------|------------------------|
> | Farmer     | 0.80            | 0.251                  |
> | Sheriff    | 1.00            | 0.177                  |
> | Nurse      | 0.80            | 0.229                  |
> | Attendant  | 0.70            | 0.128                  |
>
> | Prompt                                                                 | Interpretable ↑ | Mean Attr. Presence ↓ |
> |------------------------------------------------------------------------|-----------------|------------------------|
> | A large white building with a big clock tower at one corner.           | 1.00            | 0.083                  |
> | A couple of people carrying surf board walk on a beach.                | 0.90            | 0.194                  |
> | A batter swings the bat as the crowd watches attentively.              | 0.80            | 0.085                  |
> | A young skater is boarding inside of an empty pool.                    | 0.80            | 0.156                  |
> | A man talks on his cell phone while he surfs his computer.             | 1.00            | 0.210                  |
>
> Across all settings, interpretability remains consistently high across both WinoBias and COCO prompts, confirming that the high interpretability rate reported in Table 6 is not an artifact of a single prompt. Mean attribute presence remains low across all cases, further validating that discovered attributes are genuinely underrepresented. We will include these results in the final revision of our paper.
>
> **M2/Q2.** We apply K-means (k=50) to CLIP-ViT-L/14 embeddings of 5,000 "Doctor" images from SDXL and treat the 10 smallest clusters as minority candidates.
>
>
> | Method | Attr. Presence |
> | -------- | -------- |
> | CLIP k-means     |  0.30  |
> | RAIGen | 0.15 |
>
> Upon qualitative inspection, we observe that the majority of the clusters are not coherent minority attributes. Some contain dominant-stereotype images mixed with minority instances, or some of them were uninterpretable on an attribute level, since clustering does not always yield interpretable clusters. 3 clusters correspond to interpretable minority attributes that match with RAIGen. RAIGen additionally identifies so many other attributes, such as "surgical face mask, medical charts background," etc., that are entirely missed by clustering. These results confirm that neuron-level decomposition is necessary for this task. We will extend this analysis to more prompts and add to the final version of our paper.
>
>
> **m2/Q3.** We apologize for the ambiguity. GPT-5.2 produced all final neuron annotations reported in the paper. The Llama 4-Scout prompt shown in Appendix D.1 is the same prompt template used with GPT-5.2 for annotation,  which we inadvertently presented under the Llama 4-Scout heading. For evaluation, we use a separate ensemble of Llama 4-Scout and Qwen3-VL-8B-Instruct (Section 5.2), ensuring complete independence between annotation and evaluation models. We will clarify this distinction in the revision.
>
> **m3/Q4.** We provide the end-to-end timing breakdown for a single prompt on SDXL:
>
> | Stage | Time taken|
> | -------- | -------- |
> | Image generation + bottleneck extraction     | 55 min     |
> |MSAE training + neuron discovery | 16 min |
> | LLM annotation | 44 sec |
>
> We acknowledge that the per-prompt cost is non-trivial (discussed in Section B of the main paper). However, the dominant cost is image generation, shared by any auditing approach, including OpenBias, since large-scale sampling is unavoidable for identifying model failures. Moreover, RAIGen is a targeted auditing tool, not a per-inference module. It need not run over all prompts. For instance, an organization deploying a model for recruitment imagery would audit specific prompts like "doctor" or "CEO", and discover attributes persist for reuse in amplification or cross-model tracking. We outline a direction in our future works (Section B) in the main paper to further reduce per-prompt cost by training a shared MSAE across diverse prompt mixtures with prompt-conditioned latent selection.
>
> **m1 & m4**. We thank the reviewer for both suggestions. Due to time constraints during the rebuttal period, we were unable to conduct these experiments, but we agree they would strengthen the paper. We will include a controlled comparison with human-written descriptors (m1) and a sensitivity analysis of alternative Minority Score functional forms (m4) in the final version.

---

> > ### Author Rebuttal · Reviewer_WB4Q · 2026-04-03
> >
> > I thank the authors for the detailed rebuttal with concrete new experiments.
> >
> > M1 (interpretability generalization). My primary concern was that the
> > interpretability analysis was limited to a single prompt and model. The
> > extended results across four additional WinoBias professions and five COCO
> > prompts show consistently high interpretability rates (0.70–1.00) with low
> > attribute presence, which substantially addresses this concern. Inter-rater
> > reliability remains unreported, and these extensions are limited to SDXL,
> > but these are refinements rather than fundamental gaps.
> >
> > M2 (baseline comparison). The added CLIP k-means comparison on the Doctor
> > prompt provides useful initial evidence in favor of RAIGen (Attribute
> > Presence 0.15 vs. 0.30), although it remains limited to a single prompt.
> > I therefore view this concern as partially but meaningfully alleviated
> > rather than fully closed.
> >
> > m2 and m3 are now clearly addressed: the annotation/evaluation pipeline
> > is clarified, and the per-prompt timing breakdown is informative. m1 and
> > m4 remain as revision promises.
> >
> > Overall, the rebuttal strengthens rather than changes my prior assessment.
> > I therefore maintain my score of 4 (Weak Accept).

---

> > > ### Author Response · Authors · 2026-04-08
> > >
> > > We thank the reviewer for the thoughtful feedback. We are glad that M1, m2, and m3 are now addressed. To further resolve the remaining concern (M2), we extend the CLIP k-means baseline comparison to five additional WinoBias professions on SDXL and report the attribute presence values below:
> > >
> > >  | Profession | CLIP clustering | RAIGen |
> > > |------------|--------------|---------|
> > > | Doctor     | 0.30         | 0.15    |
> > > | Sheriff    | 0.50         | 0.18    |
> > > | Farmer     | 0.69         | 0.25    |
> > > | Attendant  | 0.47         | 0.13    |
> > > | Nurse      | 0.32         | 0.23    |
> > > | Assistant  | 0.35         | 0.15    |
> > >
> > > RAIGen consistently achieves lower attribute presence across all six professions, confirming that it identifies more strongly suppressed attributes than CLIP-based clustering. Qualitatively, we observe that k-means clusters frequently mix dominant images with minority instances, and some clusters are uninterpretable at the attribute level, making it difficult to isolate coherent minority attributes. We believe this extended comparison across six professions comprehensively addresses the remaining concern.
> > >
> > > We remain committed to including the controlled comparison with human-written descriptors (m1) and sensitivity analysis of alternative score functional forms (m4) in the revised paper.

---

### Official Review · Reviewer_wn22 · 2026-03-15

**Soundness:** 3
**Presentation:** 3
**Significance:** 3
**Originality:** 2
**Overall Recommendation:** 4
**Confidence:** 4

**Summary:**

In this paper authors proposed RAIGen, by using CLIP anchored on internal representation of the generator and a matroyshka sparse auto encoder, they introduce a rare attribute discovery method for generative models that aims to lower down the inherited biases derived from the rare attributes to the down stream synthetic datasets generated by the generative models, the authors show their methods effectiveness on both more classical unet based generators (e.g., SDXL) and also more recent DiT based generators (e.g., FLUX).

**Compliance With Llm Reviewing Policy:**

Affirmed.

**Key Questions For Authors:**

Please see the previous section.

**Limitations:**

Please see the previous section.

**Strengths And Weaknesses:**

# Strengths,
* The extent and coverage of different generators are noticeable. The paper covers both more classical unet-based diffusion models and also FLUX, which is a more recent dit generator, and this is a real strength for the generality claim.
* It is also nice that the paper verifies the toy rarity experiment across multiple seeds rather than reporting a single run. This makes the main low activation frequency corresponds to rarity claim somewhat more convincing in the controlled setting.
* although not new, core idea is interesting: using sparse autoencoders on internal diffusion representations to surface attributes that are encoded but expressed rarely is a reasonable and potentially useful direction.

# Weaknesses
* Although the authors claim the method is strictly unsupervised, I do not think this is fully validated as written. The method does not use predefined attribute labels, but it still depends on strong pretrained semantic priors, especially CLIP-style embeddings for semantic distinctiveness and llm/vlms systems for annotation and evaluation. So I think the claim should be softened or made more precise: it is label-free, but not fully free of external semantic supervision priors.

* I am also not fully convinced by the paper’s notion of minority in the conditional generation setting. Isn’t it natural to assume that if we do not specifically mention a desired attribute, the generated output will follow the distribution already present in the training data or learned model distribution? For example, if teacher in LAION-style data is skewed toward male-presenting images, should we really expect something else when prompting only “a photo of a teacher”? (as at the end of the day diffusion models are a likelihood based generators) In other words, are we observing true suppression of female teacher or just ordinary conditional imbalance? I think generative modeling is a bit different from representation learning here, and the paper needs a more precise definition of minority that takes the prompt condition c more seriously (e.g., what will happend if we add the specific prompt to the model?).

* The practical usefulness is also still somewhat unclear to me. Maybe some more practical experiments are needed. For example, does this actually help in generating better images or videos, or in some realistic auditing / intervention setting? The paper does show amplification experiments, but the overall benefit in downstream use still feels a bit abstract. Some more discussion of real use cases, maybe also in out-of-distribution generation or text-to-video settings, could help clarify why this matters in practice

* The appendix says the method is prompt-specific and currently requires a separate SAE for each auditing setup, and the experiments generate 5000 images per prompt (why 5k by the way, an ablation would be useful here). This makes me wonder how practical the method is, and also how many generations are actually needed before rarity can be estimated stably. A sample-complexity or stability study across generators would strengthen the paper a lot.

* Do the authors believe the method can be used in more mode-collapsing generators, for example GANs, or is it fundamentally tied to diffusion-model intermediate representations?

* Besides CLIP, have the authors experimented with other image-text alignment models, such as SigLIP? Since the method relies on semantic geometry, it would be useful to know how robust the results are to the choice of alignment model.

* Beyond the toy example, is there a stronger reasoning form of the minority score and especialy the distinctivenss term? At the moment Eq. 3 feels somewhat heuristic. I wonder whether a formulation more aligned with how CLIP-like models are trained would make more sense.

* a minor presentation suggestion: maybe it is best to switch the FLUX experiments into the main paper and move some of the SDXL material to the supplementary. Since the architecture-generalization claim is one of the more noticeable strengths, that result deserves more visibility.

---

> ### Author Rebuttal · Authors · 2026-03-30
>
> We thank the reviewer for the valuable feedback. We address their concerns below:
>
> **W1.** The core pipeline (MSAE, Minority Score) operates without predefined labels or categories. CLIP provides a pretrained semantic prior for distinctiveness, but rarity is mainly determined by activation frequency. LLMs/VLMs are used only for post-hoc annotation, not neuron selection or ranking, analogous to how unsupervised clustering uses external annotator interpretations. We will adopt more precise terminology in the revision and describe RAIGen as **label-free**, requiring no predefined minority categories while acknowledging pretrained priors.
>
> **W2.** We agree that likelihood-based models reflect training data, so skew under imbalance is expected. However, Friedrich et al. show diffusion models can amplify biases beyond training statistics, so the issue extends beyond conditional imbalance. We believe that documenting these imbalances is valuable regardless of source. Accordingly, we define minority w.r.t.  $P_G(a|c)$, not the data, and require $P_G(a|c) > 0$, targeting learned but underexpressed attributes. Whether due to data or amplification, the effect is the same: some attributes are systematically rare in default generations. Adding the specific attribute to the prompt yields the desired output, as confirmed in Sec. 5.4. However, this assumes prior knowledge of rare attributes. RAIGen addresses this by automating their discovery to enable informed intervention.
>
> **W3.** Beyond cross-model auditing and amplification, we highlight broader applications. 1) Targeted rare generation. Our amplification experiments already demonstrate a full discover-then-amplify pipeline. This is critical in the context of two existing approaches: (a) minority-heuristic methods amplify statistically rare content without identifying what is being amplified (as validated for Reviewer LoeD, W3.1), and (b) rare-generation methods such as R2F [1] produce high-quality outputs but require prior knowledge of which concepts are rare. Both lack a key prerequisite: identifying suppressed attributes worth amplifying. RAIGen fills this gap, and its discoveries can directly provide target specifications for pipelines like R2F, automating their manual identification step. 2) Extension to text-to-video. Although not evaluated on video, RAIGen is architecture-agnostic. Extending to video diffusion models introduces temporal axes of underrepresentation. For example, in “pedestrian crossing a street,” RAIGen may surface safety-relevant modes, such as nighttime conditions or atypical pedestrian behaviors, highlighting a promising direction for future work.
>
> [1] Park et al., Rare-to-Frequent, ICLR 2025.
>
> **W4.** Computational cost is discussed in our response to Reviewer WB4Q (m3). We also conducted a stability study on the SDXL generations required:
>
> | Samples | Interp. Neurons | Attr. Presence  |
> |-------------------|---------------------------|------------------|
> | 100              | 0.1                       | -                |
> | 1000             | 0.5                       | 0.41             |
> | 5000             | 0.8                       | 0.15             |
> | 10000            | 0.9                       | 0.14             |
>
> At 100 samples, discovery is unreliable. Performance stabilizes at 5000 samples, closely matching 10000, with diminishing returns beyond. We thus select 5000 as a cost–quality trade-off.
>
> **W5.** RAIGen is not specific to diffusion models. It requires intermediate representations that can be decomposed into interpretable features via SAEs. We believe that any generator with semantically meaningful representations is applicable, and GANs with well-studied interpretable spaces (e.g., StyleGAN style space) are therefore plausible candidates.
>
> **W6.** Following the reviewer’s suggestion, we replace CLIP ViT-L/14 with SigLIP for computing semantic distinctiveness (Eq. 3). For “A photo of a doctor,” 9/10 top minority neurons overlap, and the sole mismatch is non-interpretable under both. We attribute this stability to the nature of our distinctiveness metric, a coarse semantic contrast that is reliably captured by any vision-language model trained on large-scale image-text data. The high overlap indicates RAIGen reflects stable MSAE structure rather than embedding-specific artifacts.
>
> **W7.** CLIP’s contrastive training embeds images and text in a shared space where cosine similarity captures semantic relatedness, leading to semantically similar images clustering nearby in this space. Our distinctiveness term leverages this geometry by measuring the distance between a neuron’s activation-weighted centroid and the dataset mean in CLIP space, grounding Eq. 3 in CLIP’s learned semantics rather than an arbitrary metric. Exploring more principled formulations (e.g., information-theoretic or contrastive-objective-based) is a promising future direction.
>
> **W8.** We agree with the reviewer and will move FLUX results to the main paper.

---

> > ### Author Rebuttal · Reviewer_wn22 · 2026-04-03
> >
> > Thanks the authors for the detailed feedback and additional experiments.
> > Keeping the score.

---

> > > ### Author Response · Authors · 2026-04-08
> > >
> > > We sincerely thank the reviewer for acknowledging that the concerns have been adequately addressed and for the constructive feedback throughout the review process.

---

### Decision · Program_Chairs · 2026-04-30

**Decision:**

Accept (regular)

**Comment:**

This paper has received positive final evaluations, with four Weak Accept recommendations.

Reviewers generally appreciate the idea behind the proposed method and acknowledge the importance of unsupervised rare attribute discovery in text-to-image diffusion models. Furthermore, other identified strengths include the generality of the method, supported by experiments on different architectures including Stable Diffusion and Flux, the organization of the paper, and the originality of framing bias discovery problem as rare attribute identification.

On the other hand, the reviewers raise some concerns including limited interpretability analysis, missing comparisons with simpler rare attribute discovery strategies, limited methodological novelty, the needing of more evidence to support some claims of the paper, and the lack of relevant and competitive baselines.

In their rebuttal, the authors provided additional experiments and clarifications to the raised concerns.
All the reviewers acknowledged the effort of the authors in the provided responses, and the major concerns are considered resolved, even if some doubts regarding the needing of additional evidence to fully support broad and social usefulness remain for two reviewers.

After carefully evaluating the initial reviews, rebuttal, and the subsequent discussion between authors and reviewers, the AC agrees on the identified strengths, especially in terms of the generality of the method, and originality of the formulation, and despite sharing the remaining concerns, they are considered minors.

Therefore, the AC recommends acceptance of this paper, given that the additional clarifications and experiments are incorporated in the camera-ready.